# Automatic Data Curation for Self-Supervised Learning: A Clustering-Based Approach

**Huy V. Vo[1]  Vasil Khalidov[1]  Timothée Darcet[1,2]  Théo Moutakanni[1,3]  Nikita Smetanin[1]**
**Marc Szafraniec[1]  Hugo Touvron[1]  Camille Couprie[1]  Maxime Oquab[1]  Armand Joulin[4]**
**Hervé Jégou[1]  Patrick Labatut[1]  Piotr Bojanowski[1]**

[1]**Meta Fundamental AI Research (FAIR)**   [2]**INRIA**   [3]**Université Paris Saclay**   [4]**Google**

*Reviewed on OpenReview:* `https://openreview.net/forum?id=G7p8djzWOl`

## Abstract

Self-supervised features are the cornerstone of modern machine learning systems. They are typically pre-trained on data collections whose construction and curation typically require extensive human effort. This manual process has some limitations similar to those encountered in supervised learning, e.g., the crowd-sourced selection of data is costly and time-consuming, preventing scaling the dataset size. In this work, we consider the problem of automatic curation of high-quality datasets for self-supervised pre-training. We posit that such datasets should be large, diverse and balanced, and propose a clustering-based approach for building ones satisfying all these criteria. Our method involves successive and hierarchical applications of $k$-means on a large and diverse data repository to obtain clusters that distribute uniformly among data concepts, followed by a hierarchical, balanced sampling step from these clusters. Extensive experiments on three different data domains including web-based images, satellite images and text show that features trained on our automatically curated datasets outperform those trained on uncurated data while being on par or better than ones trained on manually curated data. Our code is publicly available at `https://github.com/facebookresearch/ssl-data-curation`.

## 1 Introduction

Self-supervised learning (SSL) is at the core of modern state-of-the-art machine learning systems. Large language models (LLMs) are pre-trained in a self-supervised way using a language modeling objective (Radford et al., 2019; Ouyang et al., 2022; Raffel et al., 2020; Touvron et al., 2023), and foundational visual encoders are trained with different flavors of contrastive learning (Richemond et al., 2020; Chen et al., 2020; Caron et al., 2021; Oquab et al., 2023). LLMs achieve outstanding performance across all conventional natural language processing tasks, such as sentiment analysis, translation, summarisation, question answering, or dialogue. For image representation, recent models achieve accuracies above 87% on ImageNet (Oquab et al., 2023), evidencing that the gap with the absolute supervised state of the art is drastically shrinking. Besides excellent performance on standard benchmarks, those models show strong out-of-distribution generalization, opening new research avenues. SSL has been successfully applied to more narrow domains, unlocking considerable model improvements, such as medical image analysis (Azizi et al., 2021; Chen et al., 2024), learning phenotypic representations of cells (Ucar et al., 2021), and canopy height estimation for forest growth monitoring (Tolan et al., 2023) to name a few.

SSL is *unsupervised* because it does not require human annotations for training the model. Because of that, SSL enables scaling both the model and data without constraints regarding data annotation. However, many previous attempts at scaling models and training data size have yielded unsatisfactory results. Large language models trained on large pools of ill-curated text corpora led to subpar performance on standard

Table 1: Effect of our data curation pipeline on three different data domains: web-based images in terms of classification accuracy or ranking mAP (for "oxf-H"), text in terms of exact match ("nq" and "tqa") or accuracy ("arc-c" or "hellaswag"), and satellite images in terms of block $R^2$ scores. Our automatic curation method leads to significant gains in benchmarks compared to the raw datasets. Best results are in bold.

| curation | web-based images | | | | | | text | | | | satellite images | | | |
|---|---|---|---|---|---|---|---|---|---|---|---|---|---|---|
| | in-val | in-A | sketch | cars | oxf-H | inat18 | arc-c | hellaswag | nq | tqa | neon | a-neon | ca | sao-paulo |
| ✗ | 82.8 | 46.9 | 54.0 | 71.7 | 14.3 | 65.9 | 35.5 | 51.9 | 19.1 | 41.3 | 0.54 | 0.34 | 0.76 | 0.41 |
| ours | **84.7** | **66.4** | **60.5** | **82.5** | **32.1** | **75.7** | **40.1** | **53.1** | **22.5** | **43.7** | **0.64** | **0.53** | **0.79** | **0.47** |

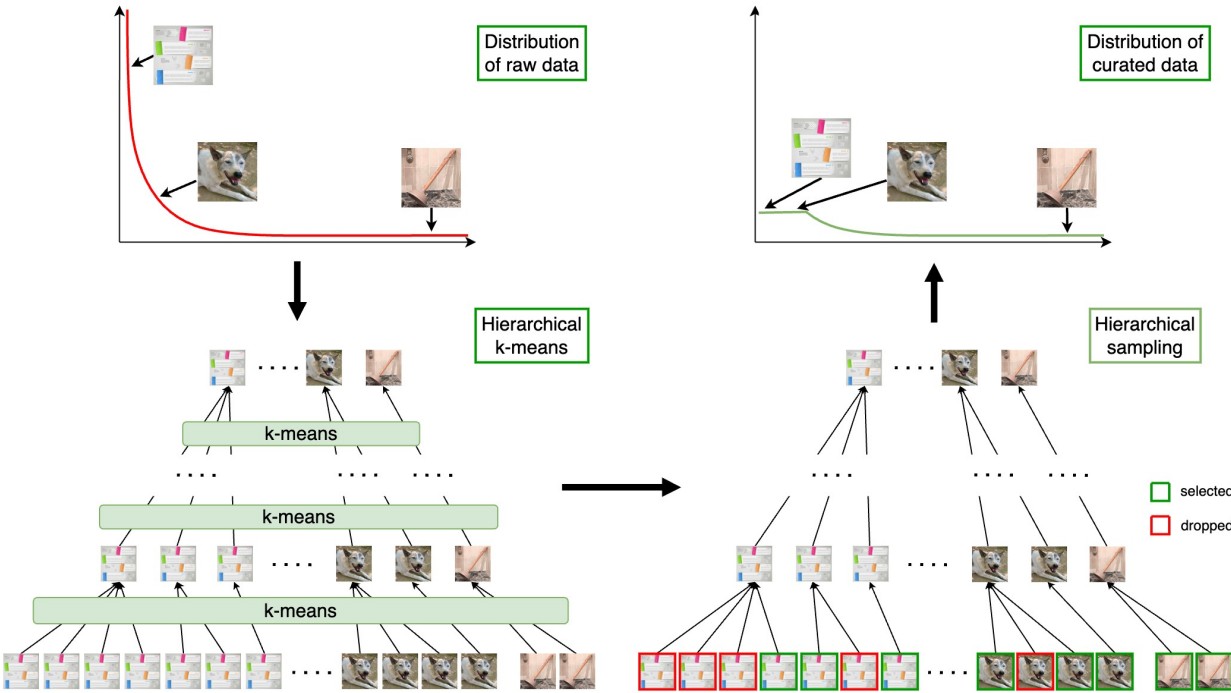

Figure 1: An overview of the data curation pipeline. Large data pool often exhibits a long-tailed distribution of concepts. On web-based images repositories, concepts such as *website* or *dog* are much more present than *plunger*. We apply hierarchical $k$-means to obtain clusters that spread uniformly over the concepts. Data points are then sampled from the clusters to form a curated dataset that has a better balance of concepts.

benchmarks (Zhang et al., 2022; Le Scao et al., 2023). Training on random collections of internet images also consistently led to a significant drop in performance (Doersch et al., 2015; Caron et al., 2019; Goyal et al., 2019; 2021; Tian et al., 2021). This poor performance is likely due to the long-tail distribution of concepts in uncurated datasets (see (Salakhutdinov et al., 2011; Zhu et al., 2014; Liu et al., 2019). As shown by Wenzek et al. (2019), web data exhibits a highly non-uniform distribution of languages, and proper language identification and filtering are required to obtain reliable monolingual text data. In image collections, specific object categories dominate the distribution and appear in many images, while others are significantly less present. Images containing a *plunger* constitute 0.1% of ImageNet but likely will be less frequent in online images. This imbalance leads to biases toward a few dominant object categories in the learned representation. We argue that balance is a necessary property of pre-trained datasets. We investigate methods for automatically rebalancing datasets with long-tail distribution.

Nonetheless there are many recent successful applications of SSL at scale. LLMs are typically trained on a carefully curated mix of data, often anchored around high-grade data sources such as Wikipedia (Touvron et al., 2023). To scale the number of tokens, raw internet data is filtered to match the language and topic distribution of Wikipedia. For foundational image models, relevant images are retrieved from a pool of

random web images base on a seed, often manually labelled dataset. This ensures a relatively good balance between visual concepts (Oquab et al., 2023). Robustness on downstream prediction tasks dramatically benefits from using large models pre-trained on large datasets. While the works mentioned above constitute strong proof points for SSL scaling, the data curation pipelines are rather *ad hoc*. In this work, we focus on the *principled* and *automatic* curation of large-scale uncurated data, which we believe will be increasingly important in future training pipelines. In order to push the limits of unsupervised pretraining, automatically designing reliable training datasets remains an open research question. As opposed to the curation procedure proposed by Oquab et al. (2023), we would like to design a generic curation algorithm agnostic to downstream tasks. A principled and generic curation algorithm allows the possibility of inferring interesting properties from completely uncurated data sources, independently of the specificities of the applications at hand.

We approach this problem from first principles and question the necessary characteristics of a good pre-training dataset. We posit that such datasets should be large, diverse, and balanced. The importance of the first two criteria has been demonstrated repeatedly (Kaplan et al., 2020; Hoffmann et al., 2022). Obtaining large and diverse data is possible by leveraging large-scale web archives of Internet (Grave et al., 2018; Wenzek et al., 2019). However, datasets assembled that way exhibit a long-tailed distribution of concepts, i.e., a few dominant concepts take up a large portion of the dataset, while others appear less frequently. This skewed distribution leads to features biased towards head concepts while ignoring those further in the tail, preventing the model from learning universal features. Therefore, we claim that the *balancing* of data is essential to avoid such biases. In our analysis, we use the term "concept" rather than "category" or "class" as the latter is often poorly defined, subjective, and depends on the context. Moreover, a data point (an image or a text paragraph) could belong to multiple such "classes". In contrast, "concept" is a more abstract term and allows us to have a more objective discussion. We do not explicitly define *concepts*, and instead let the data define it. A concept emerges as the shared content of a group of data points that are similar according to human perception. In the presence of – possibly weak – labels, a balance between concepts could be achieved by capping the number of data points corresponding to each concept (Radford et al., 2021; Dehghani et al., 2023; Xu et al., 2024). However, this is highly challenging in an unsupervised setting without metadata access.

To achieve this goal, we introduce an automatic curation technique for constructing extensive balanced datasets from an uncurated data source. From a large data pool containing a long-tail distribution of concepts, our approach aims to rebalance the data such that less frequent concepts become more prominent relative to prevalent ones. We consider a particular class of balanced datasets – ones that are sampled uniformly from the support of the underlying data distribution, and seek to build one from the data pool. Since no annotations are available, we leverage clustering-based methods to attain this goal. Given embeddings of all data points produced by a feature extractor, e.g., DINOv2 (Oquab et al., 2023) for images or SBERT (Reimers & Gurevych, 2019) for text, we introduce a hierarchical $k$-means approach that allows sampling points from a distribution that is close to the uniform distribution over the data support in the embedding space (see Fig. 1 for an overview of the proposed method). We show that self-supervised features trained on datasets curated with our method lead to large gains in benchmarks in three different domains: web-based images, text and satellite images (Tab. 1).

The rest of the paper is organized as follows. We discuss the necessary properties of pre-training datasets for self-supervised learning in Sec. 3.1, then describe the use of $k$-means or our proposed hierarchical $k$-means to build datasets with these properties in Sec. 3.2 and 3.3. We show with simulated data in Sec. 4.1 that our approach effectively flattens the data distribution, thus pulling down dense areas corresponding to redundant data and up-weighting the long-tail samples. Experiments on real-world web-based natural images are shown in Sec. 4.2 demonstrating that our approach leads to improvements on most benchmarks, in particular for robustness, out-of-distribution generalisation, and long-tailed cases. In order to assess the generality of the method beyond natural images, we apply in Sec. 4.3 our approach to text and satellite imaging data, showing significant improvements in both domains. These studies show that our method enables effectively leveraging raw data to improve self-supervised feature learning, greatly alleviating costs related to annotation and manual curation of datasets.

## 2 Related work

**Self-supervised learning** is at the core of modern machine learning. For natural language processing, language modeling is a fundamental task that is self-supervised by nature. Training neural language models started with relatively simple architectures, such as feed-forward models (Bengio et al., 2000), or plain recurrent neural networks (Elman, 1990; Hochreiter & Schmidhuber, 1997; Mikolov et al., 2010). Leveraging larger data and training larger models has opened the way to leveraging language models for representation learning (Radford et al., 2017; 2018; Devlin et al., 2019). Finetuning BERT models on the task at hand has become the standard procedure most NLP practitioners follow. Recently, pushing the language modeling paradigm to the extreme has led to astonishing progress in learning large-scale models (Chowdhery et al., 2022; Hoffmann et al., 2022; Ouyang et al., 2022; Achiam et al., 2023; Touvron et al., 2023), fundamentally changing the AI research field.

At the same time, unsupervised learning of visual features has also received much interest in computer vision over the last few years. Initially, self-supervised learning methods relied on well-tailored pretext tasks. The idea was that general visual features would emerge by training a neural network to solve these simple *ad hoc* tasks, which require the model to understand and reason about specific image properties. In parallel, several methods based on recognizing each image as its own class have been proposed (Dosovitskiy et al., 2014; Bojanowski & Joulin, 2017; Wu et al., 2018). Following this path, many alternative "general" self-supervised loss functions have been proposed, resulting in what can be referred to as Joint Embedding Architectures (LeCun, 2022). A vast body of work is dedicated to designing such losses. This includes contrastive-based methods (Oord et al., 2018; Chen et al., 2020; Hénaff et al., 2020), with variants including a momentum queue (He et al., 2020) and exploiting the nearest neighbor graph (Dwibedi et al., 2021). Along with that, some losses based on clustering (Caron et al., 2018; Asano et al., 2020; Caron et al., 2020; Assran et al., 2022), distillation (Grill et al., 2020; Caron et al., 2021), and information maximization (Zbontar et al., 2021; Bardes et al., 2022) have been proposed in the literature. This quick advance in the field led to astonishing progress of the representation power of SSL models. This work focuses on building high-quality pre-training datasets for SSL with an automatic curation approach. We evaluate our curated datasets with DINOv2 (Oquab et al., 2023), a distillation-based approach which shows successful training attempts on large, curated image datasets. Evaluating our curation pipeline with other SSL methods is beyond the scope of this work – we assume that our conclusions hold for similar training algorithms (SimCLR, MoCo, SwAV).

**Data curation.** High-quality data has been a key component in training state-of-the-art models, both for NLP and computer vision. In the context of self-supervised learning, where no metadata is required, it is still essential to leverage large volumes of high-quality data. As one of the first striking examples, the word vectors trained with word2vec (Mikolov et al., 2013) have been extremely popular with practitioners. Their quality was directly influenced by using a carefully selected dataset of more than 1B words. In order to produce high-quality word vectors in many more languages, Grave et al. (2018) have further pushed the direction of large-scale data curation. By filtering Common Crawl data, the authors managed to obtain large datasets to train reliable word vectors for 157 languages. Recently, state-of-the-art open-source LLMs (Touvron et al., 2023) also leverage this type of carefully curated large data from the web (Wenzek et al., 2019).

Most successful self-supervised visual models are trained on the curated ImageNet dataset (Deng et al., 2009) (without labels). There have been some initial attempts at training on other datasets, often created from large uncurated data sources. Doersch et al. (2015) show that self-supervised methods can be trained on uncurated visual data, and Caron et al. (2019); Goyal et al. (2019) show how it can be scaled to hundreds of millions of uncurated images. Similarly, Goyal et al. (2021; 2022a) leverage billions of random internet images to obtain high-quality self-supervised features. Asano et al. (2021) propose an ImageNet-sized dataset of uncurated images to facilitate research beyond curated images. With the goal of training on uncurated data, Tian et al. (2021) propose a solution inspired by the divide and conquer algorithm. In that setup, the uncurated dataset is split into coherent parts using clustering, and individual models are trained with SSL on those parts. A large model is obtained by distilling knowledge from the part-specific models. Oquab et al. (2023) propose to retrieve the closest images to a fixed set of datasets of interest in an unsupervised manner. This principle shares some similarities with the general idea of label propagation for semi-supervised learning, which was explored by Yalniz et al. (2019). This method shows good results, but selecting images based

on queries from specific datasets may ignore a wide range of visual concepts in Internet-based repositories. Contrary to prior work, we do not use image labels in our curation pipeline. Instead, we rely on clustering methods to select a balanced but diverse set of images for self-supervised training.

**Data pruning and active learning** seek to reduce the size of the training dataset to save computation and/or annotation cost. Data pruning finds and removes redundant or troublesome data points from the training set to improve learning. Typical approaches involves ranking data points according to some pruning metrics and remove low-ranked ones. Notable metrics include distance to prototypes (Sorscher et al., 2022), training error (Paul et al., 2021), forgetting score (Toneva et al., 2019) or influence score (Feldman & Zhang, 2020). Most data pruning methods require label information. Active learning alternates between training models and selecting the best next samples to annotate until an annotation budget cost is met (Settles, 2009a). Samples are selected so as to maximize the model's performance. Common selection strategies include choosing the most representative (Geifman & El-Yaniv, 2017; Sener & Savarese, 2018) or informative samples (Brust et al., 2019; Choi et al., 2021) or both (Zhdanov, 2019; Ash et al., 2020; Vo et al., 2022). Different from these works, we do not focus on reducing resource cost and do not require data labels, we seek instead to correct the distribution of SSL pre-training data with an automatic and unsupervised pipeline.

**Clustering** or cluster analysis aims at finding structures in data by dividing it into coherent, disjoint parts. Clustering methods can be centroid-based such as $k$-means (Arthur & Vassilvitskii, 2007; Lloyd, 1982) or mean-shift (Cheng, 1995), density-based such as DBSCAN (Ester et al., 1996; Schubert et al., 2017), statistical model-based with Gaussian Mixture Model (Yang et al., 2012) or hierarchical such as agglomerative (Defays, 1977; Sibson, 1973). It has found wide applications in various scientific fields, often used to introduce structures into data to ease further analysis. In computer vision, researchers have applied clustering for image segmentation (Achanta et al., 2012), quantization (Jégou et al., 2011) or bag-of-visual-words extraction (Lazebnik et al., 2006; Jégou et al., 2010). Our use of $k$-means clustering is close to active learning (Settles, 2009b) or data pruning (Sorscher et al., 2022) methods as part of the data selection or ranking process. Contrary to them, we do not employ $k$-means since it is sub-optimal for our purpose, and instead propose hierarchical $k$-means to sample balanced datasets. Hierarchical application of $k$-means has been considered before by Nister & Stewenius (2006) to build vocabulary trees of visual concepts. In a top-down manner, $k$-means is first used to divide the dataset into multiple clusters, then a separate $k$-means is applied onto each cluster to obtain finer clusters on which the process continues recursively. In contrast, our method builds the tree in a bottom-up manner where subsequent $k$-means is applied on the centroids obtained with the previous $k$-means. As we will see in Sec. 3, in contrary to Nister & Stewenius (2006), our approach is guaranteed to produce approximately balanced clusterings. More recently, Ma et al. (2024) also employs a two-step $k$-means clustering to obtain data clusters with different granularity for training CLIP (Ramesh et al., 2021) data experts.

## 3 Approach

### 3.1 A Criterion for Creating Pre-training Datasets

Using self-supervised learning, one can potentially train models to represent all concepts adequately. However, this is only possible if the pre-training data is large, diverse, and covers enough concepts. It has been previously shown that large and diverse datasets are essential to training *large* models which produce better embeddings than smaller counterparts (Caron et al., 2019; 2021; Ramesh et al., 2021). Large and diverse datasets have also been used in recent self-supervised learning approaches and yield better performance on downstream tasks (Ramesh et al., 2021; Oquab et al., 2023) and more robust features (Goyal et al., 2022a). They are typically obtained by crawling online data repositories and applying a heuristic for data curation.

Web-based data collections, however, often have a long-tailed distribution of concepts (Reed, 2001; Liu et al., 2022). Some concepts are dominant and take up a large portion of the dataset. In contrast, many remaining concepts appear much less frequently. This discrepancy can bias the training of the model towards the dominant concepts, or in the worst case, prevent the models from learning meaningful presentations at all. We believe that balance, defined as having roughly the same number of data points per concept, is another important criterion for self-supervised learning datasets. This criterion has yet to be thoroughly studied,

Table 2: Accuracy on ImageNet classification (Deng et al., 2009) of self-supervised features trained on ImageNet and its unbalanced variants. The degree of imbalance increases with the factor $\alpha$. The original ImageNet dataset corresponds to $\alpha = 0$.

| Imbalance factor $\alpha$ | 0.0 | 0.5 | 1.0 | 2.0 |
|---|---|---|---|---|
| Accuracy | 82.7 | 79.0 | 74.2 | 57.0 |

partly due to the widespread use of balanced seed datasets such as Wikipedia or ImageNet (Deng et al., 2009), but also the challenging nature of building a balanced dataset in an unsupervised setting. We show the importance of the *balanced* criterion with empirical results for image representations. In Tab. 2, we report the performance of models trained on datasets of varying imbalance factors. We artificially generate unbalanced variants of ImageNet by resampling this dataset such that the class sizes follow a power law with the scaling exponent $\alpha$ taken in $\{0.5, 1, 2\}$. It can be observed that more unbalanced datasets (large $\alpha$) result in worse accuracy in linear classification on ImageNet.

This observation leads us to the following *proposition*: Datasets for self-supervised learning should be large, diverse, and balanced. Data curation for SSL thus involves building datasets with all these properties. We propose to build such datasets by selecting balanced subsets of large online data repositories. As these repositories already cover a diverse set of concepts, a large balanced subset satisfies all the criteria. It is noteworthy that compared to the raw data repository, its balanced subsets down-weight head concepts in favor of making tail concepts more prominent. This could lead to a performance drop in downstream tasks involving the head concepts, as we will see in Sec. 4.2.3. However, by balancing datasets, we do not seek to improve features' performance on specific individual downstream tasks or data domains, but generally on all tasks and domains. We confirm this with empirical results in Sec. 4.2.

If all data points in the repository are associated with categorical labels, curation would simply involve sampling the same number of data points from each category. When such labels are unavailable, we can imitate this process by dividing the data pool into clusters using methods such as $k$-means (Lloyd, 1982; Arthur & Vassilvitskii, 2007) and considering the cluster index as a proxy category. We will discuss the limitations of this approach in the next section. In this work, we propose a more general approach: sampling data points from the uniform distribution over the support of the data distribution. A subset obtained that way is asymptotically balanced concept-wise if the embedding space in which the distribution is manipulated is well organized. In such space, data points that are semantically more similar lie close to each other, or in other words, the induced metric distance reflects the "semantic distance". For example, an embedding space where concepts are represented with small, non-overlapping blobs of equal size would be an ideal space. In this case, our proposed sampling approach is asymptotically equivalent to sampling the same number of points from each concept. Since access to such embeddings for web-based data is not available in practice, we approximate them with existing embeddings that are known to induce a meaningful distance function such as DINOv2 (Oquab et al., 2023) or SBert (Reimers & Gurevych, 2019).

**Problem statement.** Let $P$ be the data distribution from a source we can sample from, e.g., the Internet, and $X$ a set of samples drawn from $P$. We suppose that data are represented by vectors in $\mathbb{R}^d$ such that $X$ is an element of $\mathbb{R}^{n \times d}$. We want to select a subset $S$ of $X$ as if we directly sampled it from $U$, the uniform distribution over the support of $P$. Note that with reasonable assumptions, $U$ is well defined. Let $p$ be the density of $P$, we suppose that $P$ lives in a compact set in $\mathbb{R}^d$, i.e., its support $\Omega = \overline{\{x \mid p(x) > 0\}}$ is bounded. This assumption is reasonable since our data points are features extracted from neural networks, which are always numerically bounded and typically have small norms. The indicator function $\mathbb{1}_\Omega$ is measurable since $\Omega$ is measurable and finitely integrable thanks to the compactness assumption. We can define $U$ as the probability distribution with density $u = \frac{1}{\text{vol}(\Omega)}\mathbb{1}_\Omega$ where $\text{vol}(\Omega) = \int \mathbb{1}_\Omega$ is the volume of $\Omega$.

Next, we discuss using $k$-means clustering to sample a balanced data pool subset and its limitations. We show that we can address these limitations with a better choice of distance function or, more simply, our proposed method, which is based on a successive, hierarchical application of the $k$-means algorithm on the raw data. The centroids of clusters obtained with this method follow a distribution close to $U$. Sampling

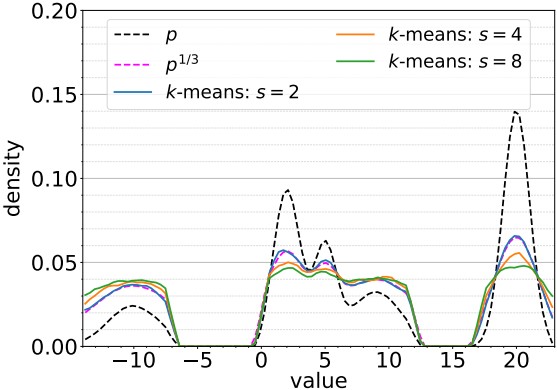

Figure 2: Normalized histograms of centroids computed by $k$-means with $d(x,y) = \|x - y\|^s$ for different values of $s$. The vanilla $k$-means centroids ($s = 2$) approximately follow the theoretical Panter and Dite formula with un-normalized density $p^{1/3}$ (Panter & Dite, 1951) with $p$ is the data distribution's density. Larger values of $s$ result in flatter distributions of centroids.

from these clusters is thus approximately equivalent to sampling directly from $U$. Finally, we discuss several data selection methods from the obtained clusters.

## 3.2 Rebalancing datasets with k-means

$K$-means clustering (Arthur & Vassilvitskii, 2007; Lloyd, 1982) is a computationally affordable technique for finding coherent structures in data. It divides data into groups such that data points in the same group are close to each other, according to some distance, while those in different groups are far away. Let $x_i \in \mathbb{R}^d$ be the embedding of data point $i$, $k$ the number of clusters, and $m_{ij}$ the binary membership variable indicating if data point $i$ belongs to cluster $j$. $K$-means seeks to find $(m_{ij})_{1 \leq i \leq n, 1 \leq j \leq K}$ that minimize the total intra-cluster distortion:

$$\sum_{j=1}^{k} \sum_{i=1}^{n} m_{ij} d(x_i, c_j), \tag{1}$$

with $c_j$ is the centroid of cluster $j$, chosen to minimize $\sum_{i=1}^{n} m_{ij} d(x_i, c_j)$, and $d$ is the squared $L_2$ distance. Note that with this choice of distance, $c_j$ has a closed form. It is the mean of all points in cluster $j$. As previously discussed, $k$-means can be employed to rebalance an uncurated dataset. One starts by dividing the dataset into multiple clusters, and taking a fixed number of images from each cluster to form a new dataset. This approach is effective only if each concept takes roughly the same number of clusters. However, it is not trivial to guarantee this condition in practice. Dominant concepts often occupy substantially more clusters than less frequent ones. For example, when applying $k$-means on a web-based image data pool, we observe that 300 out of 10,000 clusters represent "website" (see Sec. 4.2.2 and Fig. 4 for more details.).

This phenomenon is explained by looking at the objective function in Eq. (1). When a dominant concept is represented by many data points, grouping all of them in a single cluster would result in a large intra-cluster distortion. It is preferable to break this concept into multiple smaller clusters to substantially reduce the objective, and compensate for the increase in the number of clusters by grouping small, rare concepts with a slight increase in objective. As such, $k$-means tends to break large dominant visual concepts into multiple clusters. In order to illustrate this, we can consider a toy example in $\mathbb{R}$ with $k = 3$ and a dataset consisting of 5000 points equally spaced in $[0.9, 1.1]$, 2 points at $x = 2$ and 2 points at $x = 3$. Intuitively, one would choose 3 centroids at 1, 2 and 3, resulting in a distortion of 16.7. $K$-means would however break the first cluster of 5000 points in two and select centroids at 0.95, 1.05 and 2.5 to obtain a smaller distortion of 6.0.

Results from Zador (1982; 1964) provide another explanation to the phenomenon. It turns out that in $d$-dimension, the $k$-means centroids asymptotically follow the distribution with density proportional to $p^{d/(d+2)}$. See Gray & Neuhoff (1998) for a historical perspective and Fig. 2 for a 1-D simulation illustrating this

property. In high dimension, the distribution of $k$-means centroids thus depends on, and stays close to, the data distribution $P$. It means that $k$-means forms significantly more clusters in higher-density areas in the embedding space, which correspond to dominant concepts. As a consequence, it is impossible to rebalance datasets with a simple $k$-means. We will see in Sec. 4.2.2 that this is also the limitation of other clustering techniques such as Agglomerative Clustering (Defays, 1977; Sibson, 1973).

We can encourage $k$-means to form large clusters, and consequently fewer clusters in dense areas, by using another distortion function $d(x, y) = \|x - y\|^s$ with $s > 2$ (see Fig. 2). This choice of distortion keeps the cluster assignment unchanged since the closest centroid to a point according to $d$ or $L_2$ is the same. It is thus compatible with the semantic distance approximated by $L_2$. However, these new distortion functions down-weight points closer to the centroids. Having many of these points in a cluster does not substantially increase the intra-cluster distortion, reducing the impact of the cluster size. In the extreme case when $s \to \infty$, the objective only takes into account the furthest data points from the centroid in each cluster and entirely ignores the cluster size. A drawback is that the computation of a cluster's centroid given its members is no longer trivial. It can be done approximately with stochastic gradient descent, but the computation is expensive in a large-scale setting. As shown next, we can obtain the same effect with a simple successive application of the original $k$-means algorithm.

### 3.3 Rebalancing datasets with hierarchical k-means

As stated above, the centroids of $k$-means clusters follow a distribution $Q$ with un-normalized density $p^{\frac{d}{d+2}}$, which stays close to the data distribution $P$ in high dimension. However, we observe that $Q$ moves closer to $U$ than to the data distribution $P$. This is shown by the lemma below.

**Lemma 1** *Let $P$ be the probability distribution with density function $p$, $t$ a scalar in $(0, 1)$, $Q$ the probability distribution with density $q = \frac{1}{Z}p^t$ where $Z = \int p^t$, and $U$ the uniform probability distribution over the support $\Omega$ of $P$ with density $u = \frac{1}{vol(\Omega)}\mathbb{1}_\Omega$. The following inequality holds:*

$$D_{\mathrm{KL}}(Q\|U) \leq D_{\mathrm{KL}}(P\|U), \tag{2}$$

*where $D_{KL}$ denotes Kullback-Leibler divergence. Furthermore, equality happens if and only if $P = U$.*

We can prove this lemma with some simple transformations, and using the non-negativity of $D_{\mathrm{KL}}(P\|Q)$ and $D_{\mathrm{KL}}(Q\|P)$. Expanding $D_{\mathrm{KL}}(Q\|U)$ as $\int q \log q + \log \mathrm{vol}(\Omega)$ and $D_{\mathrm{KL}}(P\|U)$ as $\int p \log p + \log \mathrm{vol}(\Omega)$, we observe that Eq. (2) is equivalent to $\int q \log q \leq \int p \log p$. Thanks to the non-negativity of $D_{\mathrm{KL}}(Q\|P)$, we have $\int q \log q \geq \int q \log p$. Expand $\int q \log q$ as $t \int q \log p - \log Z$ and replace this into the previous inequality, we have $\int q \log p \leq -\frac{\log Z}{1-t}$, and consequently $\int q \log q = t \int q \log p - \log Z \leq -\frac{\log Z}{1-t}$ (*). Similarly, we can expand $\int p \log p$ as $\frac{1}{t} \int p \log q + \frac{1}{t} \log Z$ and combine with the non-negativity of $D_{\mathrm{KL}}(P\|Q)$ to have $\int p \log p \geq -\frac{\log Z}{1-t}$ (**). We deduce $\int q \log q \leq \int p \log p$ by combining (*) and (**). The equality happens if and only if $p = q$, which means $p$ is constant, or equivalently $P = U$.

We therefore propose to apply successively $k$-means to the data, in a hierarchy, to approximate $U$. In this process, we apply $k$-means $T$ times, the $t$-th $k$-means groups the set $C_{t-1}$ of the centroids of the $(t-1)$-th $k$-means into $k_t$ clusters. The first $k$-means is computed on the raw data. We call this process *hierarchical k-means* since it constructs a tree structure over the data where original data points are leaves, and inner nodes at level $t$ represent the centroids, and equivalently the clusters, obtained with the $t$-th $k$-means. The root is an imaginary point connecting nodes at level $T$. Based on the result from Zador (1982), asymptotically, the centroids of the $T$-th $k$-means follow the distribution with un-normalized density $p^{(d/(d+2))^T}$. This distribution converges to $U$ when $T \to \infty$.

In the above process, the number of input data points for $k$-means decreases exponentially after each application, which limits the number of times it can be applied. We overcome this issue with *resampling-clustering*, or simply *resampling*, steps at each level. Given clusters at level $t$, resampling-clustering involves first selecting $r_t$ points closest to the centroid from each cluster to form a subset $R$ of $C_{t-1}$. We choose $r_t$ small so that points in $R$ roughly follow the distribution of the centroids, which is closer to $U$ than the distribution of $C_{t-1}$, as shown by Lemma 1. We then apply $k$-means on $R$ instead of $C_{t-1}$ to find new $n_t$ centroids. Finally, we

---

**Algorithm 1:** Hierarchical $k$-means with resampling algorithm.

---

**Input:** Data $X \in \mathbb{R}^{n \times d}$, number of levels $T$, number of clusters per level $(k_t)_{1 \leq t \leq T}$, number of resamplings $m$, number of points resampled per cluster $(r_t)_{1 \leq t \leq T}$.

**Result:** A hierarchy of clusters on data: centroids $(C_t)_{1 \leq t \leq T}$ and clusters $((L_t^{(i)})_{1 \leq i \leq k_t})_{1 \leq t \leq T}$.

**1** **for** $t = 1$ to $T$ **do**
**2**  $\quad$ **if** $t = 1$ **then** $\mathbf{I} \leftarrow X$ **else** $\mathbf{I} \leftarrow C_{t-1}$ $\qquad\qquad\qquad\qquad\qquad$ ▷ Get input $\mathbf{I}$ of level $t$
**3**  $\quad$ $C_t \leftarrow \texttt{kmeans}(\mathbf{I}, k_t)$ $\qquad\qquad\qquad\qquad\qquad$ ▷ Find centroids $C_t$ with $k$-means
**4**  $\quad$ $L_t \leftarrow \texttt{assign}(\mathbf{I}, C_t)$ $\qquad\qquad\qquad\qquad$ ▷ Assign clusters $(L_t^{(i)})_{1 \leq i \leq n_t}$ with $k$-means
**5**  $\quad$ # resampling-clustering
**6**  $\quad$ **for** $s = 1$ to $m$ **do**
**7**  $\quad\quad$ $R \leftarrow \bigcup_{i=1}^{k_t} \texttt{resample}(L_t^{(i)}, r_t)$ $\qquad\qquad\quad$ ▷ Sample $r_t$ points from each cluster
**8**  $\quad\quad$ $C_t \leftarrow \texttt{kmeans}(R, k_t)$ $\qquad\qquad\qquad$ ▷ Find centroids based on resampled set
**9**  $\quad\quad$ $L_t \leftarrow \texttt{assign}(\mathbf{I}, C_t)$ $\qquad\qquad\quad$ ▷ Assign clusters to entire input set of level $t$
**10** $\quad$ **end**
**11** **end**

---

form a new clustering of $C_{t-1}$ by assigning points to these new centroids. Since the distribution of points in $R$ is closer to $U$ than that of points in $C_{t-1}$, we expect the distribution of the new centroids to be closer to $U$ than the distribution of the previous ones. In contrary to simple hierarchical $k$-means, resampling-clustering does not reduce the set of input points for the next $k$-means, so we can repeatedly apply this process to get closer to $U$. We use hierarchical $k$-means with resampling in our pipeline, as summarized in Algorithm 1.

**Sampling from hierarchical $k$-means.** As discussed above, the centroids in the highest level of hierarchical clustering distribute uniformly over the data support. One can build a balanced subset of the data pool by uniformly sampling a fixed number of leaves (data points) from corresponding sub-trees of the centroids. With small sub-trees with fewer leaves than needed, we take all the leaves without over-sampling. We name this sampling strategy as flat sampling. Most often, we have a target size for the subset. In this case, we find the number of points to sample from each sub-tree such that the obtained subset's size best approximates the target. Concretely, given the target size $N$ and the cluster sizes $s_j (1 \leq j \leq k)$, we find the integer $n$ that minimizes $|N - \sum_{j=1}^{k} \min(n, s_j)|$ with binary search in the interval $[0, N]$.

Instead of sampling directly from the highest-level sub-trees, we propose another strategy that samples hierarchically in a top-down manner. Given the number of points to sample from a sub-tree of level $T$, we compute the number of points to sample from its own sub-trees with the binary search above, and repeat this process down to level 1. Once we have these numbers for all clusters in level 1, we sample from them to form the balanced subset. This strategy, named hierarchical sampling, guarantees a balance among concepts represented by the sub-trees at the highest levels (such as animal, vehicle, sport, etc.) and among sub-concepts represented by internal nodes in lower levels (such as dog, spider, airplane, scooter, football, vovinam, etc.). We consider several ways to sample data points from clusters of level 1 including sampling random points ("r"), or points that are closest ("c") or furthest ("f") from their centroids. We compare the flat and hierarchical sampling, as well as "r", "c" and "f" sampling methods in Sec. 4.2.2.

**Choice of numbers of clusters.** Our analysis does not give rise to a method for choosing optimal numbers of clusters in different levels of hierarchical $k$-means. Intuitively, with the dataset's tree structure represented by the clustering, inner nodes in the low levels represent sub- or small concepts while those in higher levels represent concepts or large concepts. With this interpretation, it is reasonable to have larger clusters in lower levels (sub-concepts can contain many images) and smaller clusters in higher levels (large concepts contain several smaller concepts). Our choices of number of clusters are guided by this intuition and convenience. For example, for a 4-level hierarchical clustering on a dataset of 743M data points, we choose $k = 10M, 500k, 50k, 10k$ for the four levels, which results in clusters with average size of 70, 50, 10 and 5 clusters from the first to the fourth level respectively.

# 4 Experiments

We empirically study the proposed algorithm in several setups. We start with controlled experiments on simulated data to provide an interpretable analysis of our algorithm. Next, we perform extensive experiments by training a state-of-the-art self-supervised learning method (DINOv2 (Oquab et al., 2023)) on datasets meticulously curated from web images. Finally, we show the generality of the approach by applying the same algorithm to two other domains: the curation of text data for training large language models and the curation of satellite imagery for training a canopy height prediction model.

## 4.1 Experiments on simulated data

We first illustrate the effect of hierarchical $k$-means on simulated data in the 2-D plane. We sample 9000 points from a mixture of 3 Gaussians and the uniform distribution, bounded by the square $\Omega = [-3, 3] \times [-3, 3]$. We fit 300 clusters over this set with several configurations of hierarchical $k$-means (1, 2, 3 levels with or without resampling). We also try $k$-means with other distortion functions $||x - y||^s$ with $s \in \{4, 64, 256\}$, as well as other clustering methods such as DBSCAN (Ester et al., 1996) or Agglomerative Clustering (Defays, 1977). Using kernel density estimation, we estimate the density of the distributions of centroids obtained with these methods. We visualize the centroids along with the Voronoi diagram of clusters in Fig 3a. The figure shows that $k$-means, equivalent to 1-level hierarchical $k$-means without resampling, produces significantly more clusters in higher-density areas. The clusters are spread more evenly over the square when we use hierarchical $k$-means with more levels. Our 3-level hierarchical $k$-means with resampling yields clusters that spread almost uniformly over the support, evidenced by the almost flat density. We also observe that increasing larger exponent $s$ of the distortion function of $k$-means results in flatter density, similar to the effect of hierarchical $k$-means, but a too large value of $s$ leads to numerical instability. Additionally, it can be observed that Agglomerative Clustering has the same limitations as $k$-means while DBSCAN also fails to produce uniformly distributed clusters.

Quantitatively, we compute the Kullback-Leibler (KL) divergence between the estimated kernel density and the uniform distribution $U$ over the data support $\Omega$. The results are shown in Fig. 3b. We observe that the distribution of centroids obtained with hierarchical $k$-means gets closer to $U$ when adding more levels and resampling steps. The KL divergence of hierarchical $k$-means with three levels and resampling is close to the lower bound given by the KL divergence between the kernel density estimated on 300 random points in $\Omega$ and $U$. The KL divergence corresponding to $k$-means approaches this lower bound when the exponent $s$ increases but it stays above that of hierarchical $k$-means even with $s$ as large as 256. These results confirm our analysis in Sec. 3.

## 4.2 Self-supervised learning on web-based images

### 4.2.1 Training data, implementation details, and evaluations

We apply our curation algorithm to a pool of web-based images. It is assembled by following links from `` tags in web pages of a publicly available repository of crawled web data. We filter out URLs that point to unsafe or restricted domains and do not download images from them. Next, post-processing including de-duplication based on PCA hash, discarding images whose smallest size is smaller than 112px or greater than 512px, filtering for NSFW contents, and removing identifiable faces are applied to the downloaded images to remove harmful contents and preserve privacy. After these steps, we obtain an initial set of 1.2 billion unique images. Then, we remove near-duplicate images within this set or with respect to the test sets of the evaluations considered below with the copy detection pipeline of Pizzi et al. (2022). This results in a final data pool of 743 millions unique images.

We train a ViT-L with DINOv2[1] on ImageNet1k (Russakovsky et al., 2015) and use it as our base feature extractor. In order to run $k$-means and $k$-means++ initialization at a large scale, we implement a distributed GPU-supported version of this algorithm in PyTorch (Paszke et al., 2019). Our main run involves a 4-level hierarchical $k$-means on this image pool with 10M, 500k, 50k and 10k clusters in the first, second, third and

---

[1]https://github.com/facebookresearch/dinov2

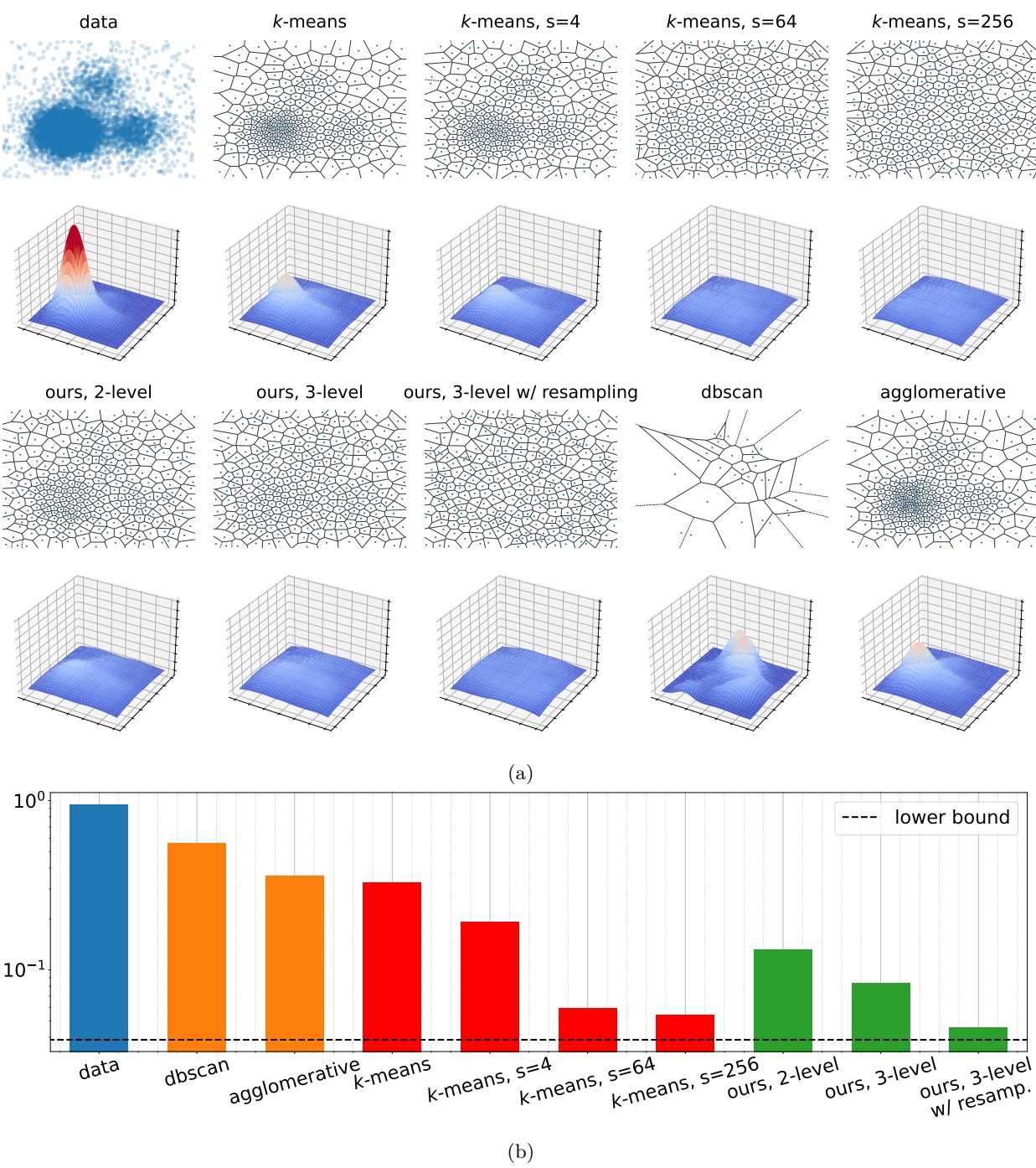

(a)

(b)

Figure 3: A visualization of clusters obtained with different clustering methods on simulated 2-dimensional data. (a) Voronoi diagrams and KDEs computed on the 2-D simulated data and the centroids of clusters obtained with $k$-means, DBSCAN (Ester et al., 1996), Agglomerative clustering (Sibson, 1973) and several variants of hierarchical $k$-means. For hierarchical $k$-means, centroids spread more uniformly with more levels and resampling steps. (b) Estimated Kullback-Leibler divergence between the uniform distribution on $\Omega = [-3, 3] \times [-3, 3]$ and the KDEs computed from the centroids.

fourth levels. Since the first level $k$-means is computationally heavy, for the sake of efficiency, we apply the resampling technique described in Sec. 3.3 10 times on the top three levels only, with the number of points

($r_t$) sampled from each cluster being half the average cluster size in each level. To form curated datasets from the hierarchical clustering, by default we use the hierarchical sampling technique presented in Sec. 3.3 with a typical target size of 100M images. To compare different pre-train datasets, we train a DINOv2-reg (Oquab et al., 2023; Darcet et al., 2024) with ViT-g for 625k iterations. For efficiency, we perform all our ablation studies with a ViT-L. We use the original training recipe from DINOv2 (Oquab et al., 2023) in all our experiments, except for a smaller learning rate of $5 \times 10^{-5}$ for ViT-g. Please refer to this work for the full set of hyper-parameters. We evaluate features pre-trained on different datasets on a wide-range of downstream benchmarks with linear probing, *without fine-tuning.*

- ImageNet classification: We report top-1 accuracy on $k$-nn and linear classification on the 1000 classes of ImageNet. Apart from the standard validation set, we also consider alternative test sets ImageNet-V2 (Recht et al., 2019) and ImageNet-ReaL (Beyer et al., 2020). These test sets have been considered before to avoid overfitting the standard validation set.

- Out-of-distribution ImageNet test sets: We report the top-1 accuracy of the linear classifier trained on ImageNet described above, on ImageNet-A (Hendrycks et al., 2021b), ImageNet-R (Hendrycks et al., 2021a), ImageNet-Sketch (Wang et al., 2019) and ObjectNet (Barbu et al., 2019). ImageNet-A contains hard examples that are incorrectly classified by trained ResNets (He et al., 2016). ImageNet-R consists of images of ImageNet categories with changes in image style, blurriness, geographical location, camera operation, etc. ImageNet-Sketch includes sketches of ImageNet classes. ObjectNet shows ImageNet objects in new viewpoints and background. These test sets are used evaluate the robustness of pre-trained features on different domains.

- Long-tailed benchmarks: We report the classification top-1 accuracy on iNaturalist2018 (Van Horn et al., 2018) and iNaturalist2021 (Van Horn et al., 2021). These datasets contain images of fine-grained natural categories such as birds, insects, plants, etc. They exhibit a highly imbalanced distribution of images among categories, presenting a challenging task.

- Retrieval: We evaluate pre-trained features on instance-level recognition of landmarks in the Oxford and Paris datasets (Philbin et al., 2007; 2008). We use the revised version of Radenović et al. (2018). We rank images based on their features' cosine similarity to the query and report the mean average precision computed based on the ranking.

- Fine-grained classification: Following Chen et al. (2020), we report top-1 classification on 12 small benchmarks. These include Aircraft (Maji et al., 2013), Caltech (Fei-Fei et al., 2004), Cars (Krause et al., 2013), CIFAR (Krizhevsky & Hinton, 2009), CUB (Berg et al., 2014), DTD (Cimpoi et al., 2014), Flowers (Nilsback & Zisserman, 2008), Food (Bossard et al., 2014), Pets (Parkhi et al., 2012), SUN (Xiao et al., 2010) and Pascal VOC (Everingham et al., 2015).

- Dense prediction: We consider three semantic segmentation benchmarks including ADE20K (Zhou et al., 2017), Cityscapes (Cordts et al., 2016) and Pascal VOC (Everingham et al., 2015), and three depth estimation benchmarks including KITTI (Geiger et al., 2013), NYU (Silberman et al., 2012) and SUN-RGBD (Song et al., 2015). We report mIoU metric on semantic segmentation and RMSE metric on depth estimation. All results were obtained by training linear heads on top of frozen patch-level features, following the process described in DINOv2 (Oquab et al., 2023). For depth estimation, we use the concatenation of 4 layers from the backbone, while we only use the last layer for semantic segmentation.

### 4.2.2 Ablation Study

We start by empirically investigating the properties of different variants of hierarchical $k$-means. First, we inspect the distribution of clusters they yield (Fig. 4). Second, we compare the performance of features trained on datasets curated with them (Tab. 3). We name curated datasets after the number of levels of the clustering from which they are generated and the intra-cluster sampling strategy. A curated dataset formed by sampling from a hierarchical clustering that has $T$ levels with "r", "c" and "f" methods are named "Tr", "Tc" and "Tf" respectively. We additionally append suffixes to indicate that flat sampling is employed instead of hierarchical sampling (see Sec. 3.3), that $k$-means is initialized randomly instead of with $k$-means++, or to explicitly specify the number of clusters in the highest level, the number of resampling steps or the type of

the base embeddings when necessary. Unless mentioned otherwise, all hierarchical $k$-means runs have 10,000 clusters in the highest level and ten resampling steps in all levels except the first one.

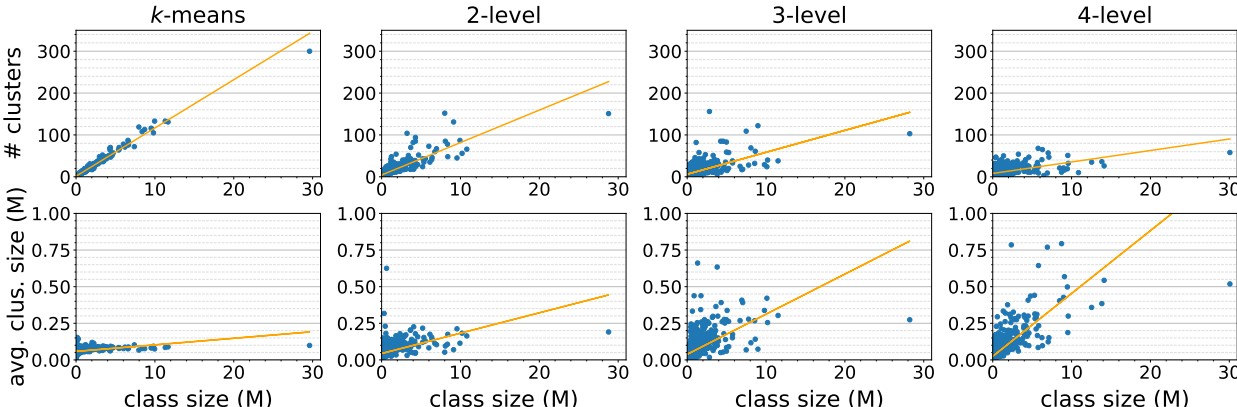

Figure 4: An investigation on the distribution of clusters of web-based images over the classes of ImageNet. The clusters are obtained with variants of hierarchical $k$-means on our data pool. For each clustering, we first assign clusters to ImageNet classes with $k$-nn, then estimate for each class their size, the number and the average size of the corresponding clusters. We show the classes' size against the number of corresponding clusters in the first row, and against the average cluster size in the second row. The straight lines that best fit the scatter points are shown in yellow. We observe that $k$-means tends to break down larger classes into more small clusters while hierarchical $k$-means with multiple levels forms fewer but larger clusters for large classes. This way, it distributes the clusters more equally among classes, regardless of their size, and enables sampling more balanced dataset from the data pool.

**Does hierarchical $k$-means lead to more balanced clusterings?** In Sec. 3 and 4.1, we provide a theoretical argument and simulations showing that our hierarchical $k$-means algorithm leads to better-balanced clusterings. Here, we investigate whether this applies to real-world data, such as our large image data pool. As discussed in Sec. 3.2, strong unbalance manifests in dominant concepts (for example, "website") being split into many small clusters. We study how the 1000 classes of ImageNet (Deng et al., 2009) relate to our clusters. We associate each cluster with one of the ImageNet categories and inspect the number of clusters and the average size of clusters representing that category.

We consider $k$-means and hierarchical $k$-means with two, three, and four levels and 10 resampling steps in each level. They result in four clusterings with 10,000 clusters at the highest level. We assign the clusters produced by each method to the ImageNet classes with $k$-nn, using the cluster centroids. We present a scatter plot of two quantities for each clustering as a function of total *class size*. These two quantities are the number of clusters associated with each class and the average size of the cluster for that class. We show the scatter plots in Fig 4. We see that $k$-means produces clusters of relatively constant size and that larger classes are broken down into more clusters. In contrast, hierarchical $k$-means with more levels can form larger clusters for large classes, leading to more equally distributed clusters among classes.

**Influence of the number of levels in hierarchical $k$-means.** We show in Sec. 3 that hierarchical $k$-means with more levels results in better balanced curated datasets, and argue that this benefits self-supervised feature learning. We empirically validate this by comparing the performance of datasets "1r", "2r", "3r" and "4r" in Tab. 3a. They are curated respectively from clusterings obtained with single-, two-, three- and four-level hierarchical $k$-means, which has 1, 11, 21 and 31 $k$-means applications. We observe that adding more levels in hierarchical $k$-means generally leads to better performance on down stream benchmarks. The biggest jump is observed when going from 1 level, which is equivalent to vanilla $k$-means, to 2 levels with significant improvements on all benchmarks. Notably, large gains are observed in robustness, long-tailed, retrieval benchmarks. Going to 3 levels leads to further gains in long-tailed and retrieval tasks. Adding

Table 3: Performance on down-stream tasks of features trained on datasets curated with different variants of our proposed hierarchical $k$-means. Datasets are named after the number of clustering levels and the sampling method within clusters, with suffixes indicating additional details in clustering configuration. All pre-training is done with ViT-L architecture. Best results are in bold, second bests are underlined. See text for more details.

(a) Influence of numbers of levels of hierarchical $k$-means.

| dataset | imagenet | | ood | | long-tailed | | retrieval |
|---|---|---|---|---|---|---|---|
| | knn | val | in-A | sketch | inat18 | inat21 | oxf-H |
| raw | 73.6 | 82.8 | 46.9 | 54.0 | 65.9 | 73.8 | 14.3 |
| 1r | 76.6 | 83.9 | 58.0 | 57.0 | 70.1 | 77.7 | 16.3 |
| 2r | 78.7 | 84.5 | 65.4 | 60.0 | 73.8 | 80.8 | 24.7 |
| 3r | 78.7 | 84.5 | 64.7 | 60.1 | **76.2** | **82.3** | 29.7 |
| 4r | **79.6** | **84.7** | **66.4** | **60.5** | 75.7 | **82.3** | **32.1** |

(b) Influence of the sampling method.

| dataset | imagenet | | ood | | long-tailed | | retrieval |
|---|---|---|---|---|---|---|---|
| | knn | val | in-A | sketch | inat18 | inat21 | oxf-H |
| 4r-flat | 78.9 | 84.4 | 64.6 | 59.2 | 74.6 | 81.5 | 18.3 |
| 4c | 79.4 | **84.7** | 64.3 | 59.4 | 75.1 | 81.6 | 29.2 |
| 4f | 75.6 | 83.4 | 54.3 | 53.5 | 68.4 | 76.4 | 16.4 |
| 4r | **79.6** | **84.7** | **66.4** | **60.5** | **75.7** | **82.3** | **32.1** |

(c) Influence of $k$-means initialization method.

| dataset | imagenet | | ood | | long-tailed | | retrieval |
|---|---|---|---|---|---|---|---|
| | knn | val | in-A | sketch | inat18 | inat21 | oxf-H |
| 4r-- | 74.0 | 83.0 | 47.7 | 53.9 | 69.0 | 76.5 | 25.4 |
| 4r | **79.6** | **84.7** | **66.4** | **60.5** | **75.7** | **82.3** | **32.1** |

(d) Sensitivity to number of clusters.

| dataset | imagenet | | ood | | long-tailed | | retrieval |
|---|---|---|---|---|---|---|---|
| | knn | val | in-A | sketch | inat18 | inat21 | oxf-H |
| 4r-20k | **80.0** | **84.8** | 65.3 | 60.2 | **76.7** | **82.7** | 31.3 |
| 4r | 79.6 | 84.7 | **66.4** | **60.5** | 75.7 | 82.3 | **32.1** |

(e) Sensitivity to number of resampling steps.

| dataset | imagenet | | ood | | long-tailed | | retrieval |
|---|---|---|---|---|---|---|---|
| | knn | val | in-A | sketch | inat18 | inat21 | oxf-H |
| 4r-0 | 77.0 | 84.4 | 63.3 | 59.0 | 74.0 | 80.9 | 26.2 |
| 4r-100 | **80.0** | **84.8** | **66.4** | 59.8 | **76.0** | **82.4** | 30.7 |
| 4r | 79.6 | 84.7 | **66.4** | **60.5** | 75.7 | 82.3 | **32.1** |

(f) Influence of the base embeddings.

| dataset | imagenet | | ood | | long-tailed | | retrieval |
|---|---|---|---|---|---|---|---|
| | knn | val | in-A | sketch | inat18 | inat21 | oxf-H |
| 4r-raw | 71.5 | 82.2 | 39.6 | 51.7 | 63.0 | 72.0 | 28.4 |
| 4r-in22k | 79.2 | **84.9** | **69.1** | **61.7** | **76.5** | **82.5** | 30.1 |
| 4r | **79.6** | 84.7 | 66.4 | 60.5 | 75.7 | 82.3 | **32.1** |

another level leads to small gains on all benchmarks except a small drop in iNaturalist2018. These results confirm the merit of our proposed hierarchical $k$-means curation method in a practical setting.

**Influence of sampling strategy.** We discuss in Sec. 3.3 two sampling strategies, the baseline flat sampling and our proposed hierarchical sampling, for forming curated datasets from a hierarchical clustering. We compare their effect in Tab. 3b ("4r" vs. "4r-flat"). It can be seen that the former outperforms the latter in all benchmarks, highlighting the importance of the balance between concepts at all levels, not just the highest one. We also compare the "r", "c" and "f" sampling methods in hierarchical sampling through the down-stream performance of features trained on "4r", "4c" and "4f" respectively. It can be observed that random sampling works best, outperforming the others on most benchmarks. It is closely followed by "c" sampling while "f" sampling is far behind. This is likely due to the fact that "f" sampling returns data points that are close to the cluster boundaries which are not guaranteed to spread uniformly in the data support. Also, "r" sampling yields a more diverse set of points than "c" sampling.

**$k$-means++ vs. random initialization.** $k$-means clustering is known to be sensitive to the centroids initialization. Randomly selecting data points as initial centroids (random) is simple and inexpensive, thus often used in large scale. It could however result in arbitrarily bad clustering with respect to the objective function (Arthur & Vassilvitskii, 2007). In our context, applying random initialization on uncurated datasets could result in important imbalance between visual concepts. On the other hand, $k$-means++ initialization (Arthur & Vassilvitskii, 2007) is known to produce more diverse clusters. It also has an approximate guarantee of optimality that enables our analysis based on the optimal solution of $k$-means in Sec. 3.3. We compare in our experiments the impact of these two initialization techniques ('4r' vs. '4r--')

on the pre-trained feature performance in Tab. 3c. We implement a distributed GPU-supported version of both techniques in PyTorch (Paszke et al., 2019). It can be seen that on all benchmarks, $k$-means++ results in features that perform significantly better than those obtained in a pipeline with random initialization for $k$-means. The former leads to features that are more robust in out-of-domain and long-tailed data, evidenced by large gains on ImageNet-A (+18.7), ImageNet-Sketch (+6.6), iNaturalist2018 (+6.7) and iNaturalist2021 (+5.8). These results stress the importance of an appropriate initialization for $k$-means, even in large scale. We initialize $k$-means with $k$-means++ by default in our experiments.

**Sensitivity to number of clusters.** As discussed in Sec. 3.3, our choices of number of clusters are guided by simplicity and the intuition that higher-level clusters should have smaller size than lower-level ones. Apart from our default clustering with $k = 10M, 500k, 50k$ and $10k$ which results in the dataset "4r", we run another 4-level hierarchical $k$-means with $k = 20M, 800k, 80k$ and $20k$. This clustering results in the dataset "4r-20k". It is observed in Tab. 3d that the two datasets leads to similar performance on average. Features pre-trained on "4r-20k" yield significantly better results on ImageNet $k$-nn and iNaturalist while those trained on "4r" perform better on ImageNet-A and Oxford retrieval. The better results with "4r-20k" on iNaturalist are likely due to the fact that more clusters in hierarchical $k$-means leads to finer, and better partition of the dataset. However, it should be noted that it also comes with a higher computational cost.

**Sensitivity to number of resampling steps.** We show in Sec. 3.3 and Fig. 3 that resampling steps allows more $k$-means applications, leading to more uniformly spread clusters over the data support. We assess its influence by comparing in Tab. 3e "4r" with "4r-0", a dataset curated from a clustering built without resampling, and "4r-100", a dataset curated from a clustering built with 100 resampling steps in levels 3 and 4. Note that we have 10 resampling steps per level in our default clustering that produces "4r". We observe that without resampling, the performance drops significantly on all benchmarks, stressing its importance. With more resampling steps, the performance further improves slightly on most benchmarks but on average, using 10 or 100 resampling steps yields comparable results.

**Influence of the base embeddings.** We conduct our main experiments with embeddings extracted from a ViT-L trained with DINOv2 (Oquab et al., 2023) on ImageNet1k. In order to investigate the influence of the embeddings, we also train a ViT-L with DINOv2 on ImageNet22k (Russakovsky et al., 2015) or our raw data pool. We then run our curation pipeline on our raw data pool with these embeddings, resulting in two new datasets "4r-raw" and "4r-in22k". It can be seen in Tab. 3f that the quality of curated datasets, as shown by the performance of features pre-trained on them, depends strongly on the type of embeddings used in the curation pipeline. Using embeddings pre-trained on the raw data pool leads to poor performance compared to embeddings pre-trained on ImageNet1k or ImageNet22k. This is likely due to the imbalanced nature of the raw data pool which leads to embeddings that are biased toward the head concepts. The distance induced by these embeddings is distorted to distinguish better samples from a few head concepts while discerning less correctly the tail concepts. Since distance between embeddings is used to define the concepts in hierarchical $k$-means, such a bad distance function leads to unsatisfactory curation results. Finally, embeddings pre-trained on ImangeNet22k lead to better performance than those trained on ImageNet1k on most benchmarks. We choose embeddings pre-trained on ImageNet1k to limit the presence of manual work in our pipeline.

**Comparisons to the base feature extractor** We compare features trained on our curated dataset "4r" and the features produced by the base extractor in Tab. 4. We observe that the model trained on "4r" performs worse than the base extractor on $k$-nn classification on ImageNet. This is not surprising since the base extractor is trained on ImageNet, thus learns to differentiate better ImageNet classes. In contrary, training features on our curated dataset leads to better performance on all other benchmarks, with large gaps on out-of-distribution, long-tailed and retrieval benchmarks.

### 4.2.3 Comparisons to other datasets.

We compare the quality of features trained on our automatically curated dataset "4r", the manually curated datasets ImageNet1k and ImageNet22k, ImageNet1k-ret – a 100M-images dataset formed by retrieving near-

Table 4: Performance of features trained on our curated dataset and those produced by the base feature extractor on common benchmarks

| dataset | imagenet | | ood | | long-tailed | | retrieval |
|---|---|---|---|---|---|---|---|
| | knn | val | in-A | sketch | inat18 | inat21 | oxf-H |
| base | **81.3** | 83.0 | 38.8 | 34.7 | 64.1 | 71.6 | 14.9 |
| 4r | 79.6 | **84.7** | **66.4** | **60.5** | **75.7** | **82.3** | **32.1** |

Table 5: Comparisons to features pre-trained on raw and manually curated datasets on downstream tasks. In all experiments, ViT-g model with DINOv2-reg (Darcet et al., 2024) is used to obtain SSL features, and we evaluate them on the benchmarks *without fine-tuning*. The terms "man." and "ret." signify manual and retrieval-based curation respectively. Best numbers are in bold, second bests are underlined.

(a) Performance on classification (accuracy) and retrieval benchmarks (mAP).

| dataset | curation | imagenet | | | | out-of-distribution | | | | long-tailed | | retrieval | | fine |
|---|---|---|---|---|---|---|---|---|---|---|---|---|---|---|
| | | knn | val | V2 | ReaL | in-A | in-R | sketch | objnet | inat18 | inat21 | oxf-H | par-H | grained |
| IN1k | man. | 72.0 | 77.7 | 65.5 | 83.6 | 21.7 | 34.8 | 24.7 | 34.2 | 42.2 | 54.2 | 11.6 | 36.9 | 78.1 |
| IN22k | man. | 83.0 | 85.9 | 77.6 | 89.4 | 74.0 | 68.9 | 55.9 | 62.9 | **81.5** | **86.0** | 32.8 | 68.6 | **91.2** |
| IN1k-ret | man. + ret. | **83.4** | **86.1** | **78.6** | **89.6** | 75.3 | **79.1** | 62.5 | 65.1 | 75.3 | 82.4 | 24.8 | 65.7 | 90.7 |
| raw | ✗ | 78.0 | 85.0 | 75.9 | 88.7 | 65.8 | 74.2 | 59.9 | 67.1 | 72.2 | 79.6 | **35.8** | 75.6 | 89.7 |
| 4r | ours | 81.5 | 85.7 | 78.0 | 89.2 | **75.4** | 79.0 | **64.1** | **69.3** | 80.6 | 85.5 | 33.2 | **79.5** | 90.9 |

(b) Performance on semantic segmentation (mIoU) and depth estimation (RMSE) benchmarks.

| dataset | curation | dense prediction | | | | | | | |
|---|---|---|---|---|---|---|---|---|---|
| | | segmentation | | | | depth ↓ | | | |
| | | ade20k | voc | cityscapes | avg | kitti | nyu | sun-rgbd | avg |
| IN1k | man. | 0.398 | 0.799 | 0.656 | 0.618 | 2.905 | 0.417 | 0.472 | 1.265 |
| IN22k | man. | 0.481 | 0.830 | 0.688 | 0.666 | 2.642 | 0.329 | 0.369 | 1.113 |
| IN1k-ret | man. + ret. | 0.468 | 0.823 | 0.687 | 0.659 | 2.703 | 0.319 | 0.371 | 1.131 |
| raw | ✗ | **0.500** | **0.837** | **0.701** | **0.679** | **2.556** | **0.312** | **0.361** | **1.076** |
| 4r | ours | 0.489 | 0.828 | 0.695 | 0.671 | 2.560 | 0.335 | 0.371 | 1.089 |

(c) Evaluation on the *fairness* of pre-trained features.

| dataset | curation | fairness | | | | | | | | |
|---|---|---|---|---|---|---|---|---|---|---|
| | | income bucket | | | | regions | | | | |
| | | low | medium | high | *rel. gap* (%) | africa | asia | americas | europe | *rel. gap* (%) |
| IN1k | man. | 48.6 | 68.3 | 79.1 | *38.6* | 55.4 | 66.3 | 72.9 | 80.0 | *30.8* |
| IN22k | man. | 65.1 | 82.6 | 89.4 | *27.2* | 72.1 | 80.3 | **86.6** | 89.2 | *19.2* |
| IN1k-ret | man. + ret. | 64.6 | 81.9 | 89.5 | *27.8* | 71.2 | 79.9 | 85.7 | 89.4 | *20.4* |
| raw | ✗ | 65.4 | 82.8 | **89.8** | *27.2* | 72.3 | 80.7 | 86.2 | **89.8** | *19.5* |
| 4r | ours | **66.7** | **82.9** | 89.7 | ***25.6*** | **72.7** | **81.3** | 86.5 | 89.0 | ***18.3*** |

est neighbors of ImageNet1k images in our data pool, and the raw data pool in Tab. 5. Architecture ViT-g is used in all these experiments.

**Comparisons on classification and retrieval benchmarks.** We show the performance of the pre-trained features on ImageNet, out-of-distribution, long-tailed, retrieval and fine-grained benchmarks in Tab. 5a. On all benchmarks, except for Oxford retrieval (Radenović et al., 2018), features trained on our curated dataset "4r" significantly outperform those trained on the raw data pool. The gap is notably large on out-of-distribution and long-tailed benchmarks, showing that our curation method leads to more robust

features. On standard ImageNet and fine-grained benchmarks, curation also yields significant improvement, confirming its merit. Finally, although the raw data pool leads to better performance than the curated dataset on Oxford retrieval, the latter produces better features on Paris retrieval benchmark.

Compared to features trained on ImageNet22k, features trained on "4r" perform slightly worse on ImageNet $k$-nn but the two are on par on other validation sets (val, V2 and ReaL). This is significant because ImageNet22k contains ImageNet1k and is curated with significant human effort while our dataset is obtained with an automatic pipeline. On iNaturalist (Van Horn et al., 2018) and fine-grained benchmarks, ImageNet22k still leads to slightly better performance, but on out-of-distribution and retrieval benchmarks "4r" results in much better results, with large gaps in ImageNet-R (Hendrycks & Dietterich, 2019), ImageNet-Sketch (Hendrycks et al., 2021a), ObjectNet (Barbu et al., 2019) and Paris retrieval (Radenović et al., 2018). This demonstrates that SSL training on our curated dataset produces features that are more robust.

Among the pre-training datasets, ImageNet1k-ret yields the best performing features on ImageNet1k classification, but this is unsurprising since this dataset is centered around ImageNet. This skewness toward Imagenet hinders the features' ability to generalize to other domains. Indeed, significant performance gaps are observed on ImageNet-Sketch, ObjectNet, iNaturalist, Oxford and Paris compared to features trained on "4r". These results highlight the limitation on generalizability of retrieval-based curation methods. Finally, we observe that ImageNet1k leads to poor results on all benchmarks. It is likely due to its small size with respect to a high-capacity model as ViT-g. This confirms again the need for large SSL pre-training datasets.

In order to assess the sensitivity of the performance on downstream tasks to different sampled curated datasets, we have generated two other "4r" datasets and run the evaluation pipeline on them. We compute the standard deviation of the performance on the benchmarks and observe that most benchmarks are relatively insensitive to the hierarchical sampling process. The standard deviation of the performance is smaller than 0.1 on ImageNet knn and linear classification, smaller than 0.3 on alternative ImageNet test sets (V2, ReaL and OOD benchmarks), smaller than 1.0 on long-tailed benchmarks (0.9 on iNaturalist2018 and 0.5 on iNaturalit2021) and smaller than 0.2 on fine-grained benchmarks. These numbers are significantly smaller than the performance gap between "4r" and "raw" on the corresponding benchmarks. The only exceptions are retrieval benchmarks which turn out to be very noisy with the standard deviation of the performance is 4.5 and 4.9 on Oxford and Paris datasets respectively.

**Dense prediction tasks.** We present the performance of pre-trained features on semantic segmentation and depth estimation in Tab. 5b. It can be seen that the raw data pool yields slightly better performance than the curated dataset. This is likely due to the distribution of our data pool. When looking at the image clusters, we observe that among the largest 50 clusters, there are 9 clusters depicting "bedroom", "indoor scene" or "building" concepts which are relevant in benchmarks such as ADE20K (Zhou et al., 2017), NYU (Silberman et al., 2012) or SUN-RGBD (Song et al., 2015). These clusters contain more than 35 millions images in total, taking up 5% of our data pool. With hierarchical sampling, around only 600 thousands of them are retained in the curated dataset, which corresponds to only 0.6% of its size. As a result, features trained on the raw data pool represent better the above concepts and obtain better performance on these benchmarks. It is however noteworthy that the performance drop caused by curation here is very small compared to the gains achieved in other benchmarks. We also observe that on average, "4r" leads to significantly better results than manually or retrieval-based curated datasets. Finally, similar to other benchmarks, the features' performance on these dense tasks are fairly insensitive to the hierarchical sampling process, with standard deviation of 0.002 and 0.006 observed for the performance of "4r" on semantic segmentation and depth estimation respectively.

**Fairness across geographic regions.** Following Goyal et al. (2022b) and Oquab et al. (2023), we evaluate features fairness on Dollar Street dataset (De Vries et al., 2019). This dataset contains images depicting various objects from 289 households from 54 countries. The model is trained to recognise 94 visually varying concepts among households based on their income level and geographical regions. Results in Tab. 5c show that features pre-trained on large datasets yield a narrower gap among income levels and regions compared to those pre-trained on ImageNet1k. Training on our curated dataset also leads to smaller gap than training on the manually curated ImageNet22k, the retrieval-based curated dataset and the raw data pool. However,

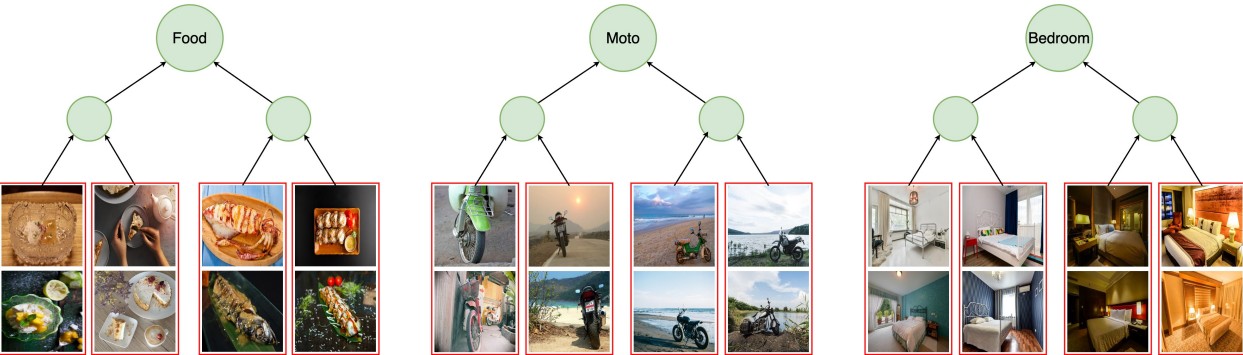

Figure 5: Hierarchy of clusters obtained when applying our proposed hierarchical $k$-means on web-based images. We show here clusters in levels 1, 2 and 3 representing *food*, *motorbike* and *bedroom* concepts. Red rectangles show clusters in level 1 with 2 representative images.

Table 6: Performance on common large language models benchmarks of models trained on raw and curated CCNet datasets. On all datasets, we train a 7B-parameters model for an equivalent of 210B seen tokens. We see that on both CCNET variants, our curation method leads to improved performance.

| dataset | arc-c | hellaswag | nq | tqa | piqa | siqa |
|---|---|---|---|---|---|---|
| ccnet-1 | 37.6 | 51.4 | 27.5 | 53.1 | 75.3 | 42.9 |
| curated ccnet-1 | **40.8** | **52.7** | **28.0** | **54.1** | **76.1** | **43.3** |
| ccnet-2 | 35.5 | 51.9 | 19.1 | 41.3 | 76.6 | 42.1 |
| curated ccnet-2 | **40.1** | **53.1** | **22.5** | **43.7** | **78.2** | **42.5** |

the relative gap among income levels (25.6%) and regions (18.3%) is still significant. This is possibly due to the limitation of our data pool. Performing curation on a much larger and more diverse raw data pool would potentially lead to better fairness in SSL features.

### 4.2.4 Qualitative evaluation of our curation method

We illustrate in Fig. 5 a sample from the hierarchy of concepts obtained on the web-based images pool. We show three clusters in level 3 that represent *food*, *motorbike* and *bedroom* concepts. Red rectangles represent clusters in level 1, each of which is illustrated with two representative images. It can be seen that the clusters are coherent. Clusters in lower levels represent finer concepts such as sub-category of objects (desert or main dish for food), style of objects (beds with different styles), background (motorbike at the sea or by the lake), viewpoint (front view or back view of motorbikes) or illumination (dark or light). The balance of these concepts and sub-concepts in curated datasets is important, as shown in the experiments above.

### 4.3 Application to other domains

The method presented in this paper is general and agnostic to the downstream task at hand. We can apply our algorithm as long as we can compute good features for the raw training data. In this section, we demonstrate the robustness of our approach by successfully applying the same method to two other tasks. First, we study the training of large language models on large-scale web-based text corpora. Second, we investigate data curation for training representations of satellite images.

### 4.3.1 Large Language Model Training

It has been repeatedly shown that large language models (LLMs) require a large amount of data for training. For example, we see a continuous improvement when training models on more tokens (see Llama 1 & 2 (Touvron et al., 2023)). Hoffmann et al. (2022) shows that using more tokens significantly improves the model's quality, irrespective of the parameter count. However, the data quality plays an essential and ill-

studied role. The first installment of Llama (Touvron et al., 2023) was trained on a mix of datasets, some of which are high-quality datasets from narrow domains while most of the training data was a variant of CCNET (Wenzek et al., 2019), a heuristic Wikipedia-based curation applied to text from Common Crawl. We investigate the effectiveness of our automatic method for curating LLM pre-training data.

To this end, we apply our curation pipeline to two text pools based on Common Crawl. The first data pool ("ccnet-1") is obtained by following Touvron et al. (2023), which employs the pipeline of Wenzek et al. (2019) followed by a Wikipedia-based filter. This dataset of 641M documents has already been curated, so the data distribution is skewed towards Wikipedia. We obtain our second data pool ("ccnet-2") by running the same pipeline, without the Wikipedia-based filtering from LLaMa and the Language Model filtering stage from Wenzek et al. (2019). Doing so keeps the original data distribution closer to the raw Common Crawl than Wikipedia. This dataset is more "raw" than "ccnet-1" and has 789M documents.

We use the `all-mpnet-base-v2` model from SBERT (Reimers & Gurevych, 2019) to represent documents. We apply 3-level hierarchical $k$-means with 10M, 500k and 50k clusters in the three levels respectively, and sample 200M documents to form curated datasets on both data pools. On each data pool and curated dataset, we train a language model with 7B parameters on a schedule for 210B tokens following Touvron et al. (2023). After training the model, we evaluate it on several tasks. We consider benchmarks including 0-shot evaluation on *common sense reasoning* tasks such as PIQA (Bisk et al., 2020), SIQA (Sap et al., 2019), Arc-challenge (Clark et al., 2018) and Hellaswag (Zellers et al., 2019), as well as 5-shot evaluation on *world knowledge* tasks such as NQ (Kwiatkowski et al., 2019) and TrivialQA (Joshi et al., 2017). We report the accuracy metric on common sense reasoning benchmarks, while on world knowledge tasks, we report *exact match* metric. The downstream performance on these benchmarks are shown in Tab. 6.

It can be seen that our curation method significantly improves performance on all benchmarks, both for ccnet-1 and ccnet-2, with large gains on the Arc-challenge and NQ datasets. It is noteworthy that our automatic curation pipeline manages to improve a data pool that was already curated. Indeed, "ccnet-1" was filtered to discard documents that would fall too far away from the Wikipedia distribution. This consistent improvement over ccnet-1 is likely due to a better balance of concepts brought by our method, an aspect often overlooked in current data pipelines.

### 4.3.2 Applications to satellite images

Tolan et al. (2023) presents an interesting application of self-supervised learning to the problem of tree canopy height estimation from satellite imagery. This work aims to build a high-accuracy map of tree height at a global scale. Such maps are helpful to monitor forest growth more efficiently and transparently. They propose a two-step approach. First, a backbone is trained using DINOv2 (Oquab et al., 2023) on a large-scale dataset of satellite images. Then, a supervised decoder is trained on top of it using smaller, high-quality annotated data. They use a pre-training dataset of 18 million $256 \times 256$ patches of satellite imagery of about 0.5-meter resolution. The images were sampled in areas where height measurements were available from the GEDI satellite, mainly selected from samples containing vegetation. The decoder is borrowed from Dense Prediction Transformer (Ranftl et al., 2021). It is trained using satellite images paired with ground truth canopy height maps from the NEON (National Ecological Observatory Network (NEON), 2022) dataset. This data covers several US regions. The canopy height estimator is then evaluated on four test sets. They include the NEON test set, which contains images from sites not present in the decoder's training data, the California Brande dataset (Brande, 2021), the Sao Paulo dataset (dos Santos et al., 2019), which contains much higher trees than those in NEON, and the Aerial NEON test set which contain images acquired by drones instead of satellites.

Table 7: Performance of Tolan et al. (2023) on canopy height benchmarks when using backbones pre-trained on raw or curated dataset of satellite images.

| dataset | neon | | ca brande | | sao paulo | | aerial neon | | avg | |
|---|---|---|---|---|---|---|---|---|---|---|
| | MAE ↓ | r2 | MAE ↓ | r2 | MAE ↓ | r2 | MAE ↓ | r2 | MAE ↓ | r2 |
| raw | 3.1 | 0.54 | **0.6** | 0.76 | 5.2 | 0.41 | 3.3 | 0.34 | 3.0 | 0.51 |
| curated | **2.9** | **0.64** | **0.6** | **0.79** | **5.0** | **0.47** | **3.1** | **0.53** | **2.9** | **0.61** |

For our experiments, we build a raw pool of 18 million images in a similar manner to Tolan et al. (2023). On this data pool, we apply a 3-level hierarchical $k$-means with 500k, 50k and 10k clusters in the first, second and third level. We then sample a curated dataset of 9 million images. We use DINOv2-reg ViT-L (Darcet et al., 2024) embeddings trained on the raw data pool to represent the images. We then train a DINOv2-reg ViT-L on both the curated dataset and the raw data pool, and evaluate the canopy height estimators trained with these two backbones. We follow the same evaluation protocol as Tolan et al. (2023) and report the Mean Average Error (MAE) and block $R^2$ (r2) metrics on the test sets. We summarize the results in Tab. 7. Training the backbone of our curated dataset leads to significant improvements on all benchmarks, with a relative improvement of 20% in the r2 metric on average. The difference in r2 is the largest on the aerial neon test set, which is the most out-of-distribution set - the imaging technology is different (airborne versus satellite). Our results demonstrate the potential of our curation pipeline to improve learning systems in domains where large-scale, high-quality curated datasets are rare or unavailable.

## 5 Conclusions

We have presented an automatic data curation pipeline that produces large, diverse, and balanced training datasets for self-supervised feature learning. Our method involves a successive application of $k$-means clustering on raw datasets, coupled with resampling-clustering steps that improve the distribution of $k$-means centroids. This procedure results in clusters that spread more uniformly among concepts. Through extensive experiments, we have demonstrated that our pipeline enables the learning of effective features in three different data domains including web-based images, satellite imagery, and text. Our pipeline leads to more robust features than those trained on manually curated datasets when applied to web-based images. These features also perform well in a broader range of tasks than those trained on datasets curated using retrieval.

Although our curated datasets yield significantly better features than raw datasets or ImageNet1k, they are still slightly outperformed by ImageNet22k on certain benchmarks such as ImageNet-1k, fine-grained classification datasets, and iNaturalist. However, it is noteworthy that ImageNet22k was curated with significantly more human effort than ImageNet1k. These evaluation datasets are very correlated with the ImageNet benchmark, which has influenced computer vision benchmarking for more than a decade. Moreover, our curated dataset still bests it on the critical robustness tests (ImageNet Adversarial, Rendition, and Sketch).

Our method leads to models that perform significantly better than those trained on raw data for text and satellite images. These results confirm the importance of data curation for self-supervised feature learning and the merit of our approach. Applying hierarchical $k$-means is not confined to the self-supervised learning context. It should be considered in place of vanilla $k$-means in tasks necessitating diverse and representative data sets, such as active learning or data pruning. Future work would point in this direction.

**Limitations.** First, our work proposes three desired properties of pre-training datasets. However, other factors are not taken into account. This includes subjective and hard-to-estimate factors such as the quality of individual data points. Second, in our experiments on web-based images, we still rely on features pre-trained using SSL on a manually assembled dataset (ImageNet-1k). Further investigations are necessary to remove this manual component from our pipeline. Finally, leveraging drastically larger image pools would further improve our performance. We leave this scaling exercise for future work.

**Statement of Broader Impact.** Automated dataset construction generally poses the risk of reinforcing biases and breaching privacy. In our work, we mitigate these concerns with several safety measures. For instance, we used strong models to detect and blur all human faces in our web-based image data pool. Furthermore, our work aims to alleviate the bias due to over-representing some concepts in random internet images, leading to better fairness in downstream tasks (Tab. 5c). At the same time, practitioners could tailor parametric curation methods for specific goals. If fairness evaluations such as those in Sec. 4.2.3 are set up, one can monitor the downstream performance along with fairness indicators to choose optimal data. The end user creating the dataset should probe for fairness issues.

## Acknowledgements

We would like to thank the M2C2 team at Meta FAIR for preparing the web-based image data pool.

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
