# OpenReview forum: "Automatic Data Curation for Self-Supervised Learning: A Clustering-Based Approach"
_TMLR — Accepted by TMLR_

### Review · Reviewer_fAds · 2024-06-18

**Summary Of Contributions:**

The paper proposes a clustering approach to balance data for training ML models. The authors describe their method and evaluate it empirically, showing improvements over other methods. In addition, they analyze the ways in which the balancing is achieved, demonstrating theoretical and empirical advantages over other methods.

**Audience:**

Yes

**Claims And Evidence:**

Yes

**Requested Changes:**

See above.

**Strengths And Weaknesses:**

The paper is well-written, tackles an interesting problem that is under-researched, and present an interesting solution that seems to work well in practice. Both the empirical evaluation and analysis of the method are comprehensive and convincing, also highlighting areas where the proposed method does not beat others.

My only complaint is that the empirical evaluation does not establish any confidence intervals for the measured performances, e.g. through repeated resampling. As in some cases the performance numbers are quite close, this would help to judge the significance of the proposed method. I realize that this may not be possible because the experimental effort might be too high, but it would help if the authors could discuss in more detail to what extent random splits into train/test affect the results, in particular with respect to long-tail distributions that may look very different when randomly split.

---

> ### Author Response · Authors · 2024-07-10
> **Answers to reviewer fAds**
>
> We would like to thank reviewer fAds for the valuable remarks and suggestions, and provide answers to the reviewer's questions below. We have also updated the manuscript and highlighted the text related to our answers in blue.
>
> *Q. My only complaint is that the empirical evaluation does not establish any confidence intervals for the measured performances, e.g. through repeated resampling. As in some cases the performance numbers are quite close, this would help to judge the significance of the proposed method. I realize that this may not be possible because the experimental effort might be too high, but it would help if the authors could discuss in more detail to what extent random splits into train/test affect the results, in particular with respect to long-tail distributions that may look very different when randomly split.*
>
> We would like to thank the reviewer for the suggestion. Regarding our evaluation, we follow the protocol of DINOv2, using the standard train/test split of the benchmarks, to ensure fair comparisons in our experiments. Following the reviewer’s other suggestion, we have measured the variation of the performance on downstream tasks when sampling different curated datasets from the hierarchy of clusters. We have generated 3 different “4r” datasets and run the experiments on them separately. The results show that most benchmarks are relatively insensitive to the hierarchical sampling process. The standard deviation of the performance is smaller than 0.1 on ImageNet knn and linear classification, less than 0.3 on alternative ImageNet test sets (V2, ReaL and OOD benchmarks), less than 1.0 on long-tailed benchmarks (0.9 on iNaturalist2018 and 0.5 on iNaturalit2021) and less than 0.2 on fine-grained benchmarks. On dense tasks, it is on average 0.002 and 0.006 on semantic segmentation and depth estimation respectively. These numbers are significantly smaller than the performance gap between “4r” and “raw” on the corresponding benchmarks. The only exceptions are retrieval benchmarks which turn out to be very noisy with the standard deviation of the performance is 4.5 and 4.9 on Oxford and Paris datasets respectively.

---

### Review · Reviewer_6tUV · 2024-06-28

**Summary Of Contributions:**

The paper introduces a clustering-based methodology for automatically curating high-quality, balanced datasets for self-supervised learning. The method uses hierarchical k-means clustering to overcome traditional challenges associated with manual dataset curation, such as high labor costs and scalability issues. Validated across diverse data domains—including web images, satellite imagery, and text—the approach demonstrates its broad applicability and shows that automatically curated datasets can match or even exceed the performance of manually curated datasets in various benchmarks.

**Audience:**

Yes

**Claims And Evidence:**

Yes

**Requested Changes:**

See weaknesses above.

**Strengths And Weaknesses:**

Strengths
* The paper proposes a novel clustering method for curating a balanced dataset for self-supervised learning which could be an important framework to reduce bias.
* The paper validates the algorithm on different domains (image, text), showing its broad applicability.
* The paper is well formatted, and the writings are clear.

Weaknesses
* The experiment sections lack hyperparameters used for the experiments (e.g., learning rate, batch size, optimizer used)
* It is not clear how the linear probing was done for the image classification tasks. Is the linear probing just a single layer on top of the trained self-supervised feature extractor?
* In Table 5, it would be helpful to state what ‘man.’ and ‘ret.’ is in the caption.
* Table 5-(c) should have the relative gap among different groups.
* For the classification tasks, I assume the paper used Cross Entropy loss. Considering the performance implications for datasets with long-tailed distributions, how might the results differ if an alternative loss function, such as Focal loss, known to be effective for imbalanced data, is used?

---

> ### Author Response · Authors · 2024-07-10
> **Answers to reviewer 6tUV**
>
> We would like to thank reviewer 6tUV for the valuable remarks and suggestions, and provide answers to the reviewer's questions below. We have also updated the manuscript and highlighted the text related to our answers in blue.
>
> *Q. The experiment sections lack hyperparameters used for the experiments (e.g., learning rate, batch size, optimizer used)*
>
> We use the original training recipe from DINOv2 in all our experiments, except for a smaller learning rate of $5 \times 10^{-5}$ for ViT-g. We have clarified this in the updated paper.
>
> *Q. It is not clear how the linear probing was done for the image classification tasks. Is the linear probing just a single layer on top of the trained self-supervised feature extractor?*
>
> Following DINOv2, we employ the standard linear probing protocol with only a single layer on top of the self-supervised feature extractor.
>
> *Q. In Table 5, it would be helpful to state what ‘man.’ and ‘ret.’ is in the caption.*
>
> We have clarified these terms in the caption of Tab. 5.
>
> *Q. Table 5-(c) should have the relative gap among different groups.*
>
> We have added the relative gaps in Tab. 5c.
>
> *Q. For the classification tasks, I assume the paper used Cross Entropy loss. Considering the performance implications for datasets with long-tailed distributions, how might the results differ if an alternative loss function, such as Focal loss, known to be effective for imbalanced data, is used?*
>
> Following the reviewer’s suggestion, we have tested the Focal loss in our evaluation on long-tailed benchmarks iNaturalist18 and iNaturalist21. However, we observe that using Focal loss instead of Cross Entropy loss leads to an absolute drop of 5.9% and 4.5% in accuracy in the two datasets respectively. It is likely that Focal loss is only relevant when learning both the feature extractor and the classifier, which is the case in the long-tailed learning literature, than in our setting when linearly separable features are already learned.  Nevertheless, we also observe that the drops when using features trained on the raw dataset are bigger (9.1% and 7.7% resp.). Therefore, our conclusion remains the same.

---

### Review · Reviewer_Q5bR · 2024-06-29

**Summary Of Contributions:**

The paper proposes an automated and principled framework for curating high-quality datasets for self-supervised pre-training. The authors argue that good pre-training sets for self-supervised learning must possess three characteristics: they should be large, diverse and balanced. This work primarily focuses on the third desired property, i.e., how to balance the data in such pre-training sets using an automated pipeline.  The idea is that concepts usually follow a long-tail distribution; the goal is to then rebalance the data so that the various concepts appear almost uniformly. The authors propose to make use of hierarchical k-means on the embeddings generated by standard feature extractors. The framework consists in first applying k-means successively and hierarchically, which provably results in high-level clusters whose centroids distribute uniformly among the various concepts. In the second phase, data points are sampled in a balanced and hierarchical manner from the constructed hierarchy of clusters. The authors conduct an extensive experimental evaluation on image data, but also investigate the potential of their framework with text data, and satellite imagery. They show that they their pipeline can generally learn more robust features that perform well on tasks different than those that they were trained on. Furthermore, a detailed ablation study confirms the various design choices of the proposed framework.

**Audience:**

Yes

**Broader Impact Concerns:**

The authors have sufficiently addressed the broader impact of their work in a dedicated paragraph.

**Claims And Evidence:**

Yes

**Requested Changes:**

There are many things to like in this work, but I feel several adjustments should be made to accommodate for the weaknesses that I previously mentioned (see also detailed discussion in Weaknesses section above).
1. Can the authors provide stronger evidence that balanced datasets are always preferable for self-supervision? If not, instead of making such a big and universal claim, could the authors elucidate those settings where balanced pre-training datasets are beneficial? For instance, in the context of OOD classification, it is reasonable to claim that balanced datasets would be preferrable. Instead of treating the claim as obvious, it would be helpful if the authors pinpointed various conditions under which balanced pre-training datasets can improve performance (with reasonable certainty), if universal arguments are much harder to write down clearly.
2. Can the authors provide any empirical data with different distortion functions (with $s>2$)? If computational overhead is a problem, what if they simply experimented with smaller synthetic datasets? This  many not be critical, but I feel it would considerably strengthen the work, if the authors could show that alternative frameworks (e.g., based on k-medoids) perform worse than the proposed hierarchical k-means.
3. Can the authors elucidate the role of embeddings? I feel the discussion in Table 3f is quite elementary. I think it is fair to say that the performance of the framework critically depends on the embeddings. How so? For instance, why 4r-raw performs worse than 4r, even though the former should cover a great deal of concepts in the embedding space? Maybe one would have expected that 4r-raw would be able to sample uniformly from a very large concept space, but in practice it performs worse. Is this expected and how exactly?
4. In the image experiments, raw performs very well on dense tasks and even on the fairness criteria. Does this suggest that the proposed framework is mostly expected to shine in OOD and long-tailed tasks, but not more generally? Some discussion there could be quite illuminating.

Trivial:
- On page 8, $\sum_j^k$  should read $\sum_{j=1}^k$.
- On page 8, sub-tress should read sub-trees.
- On page 14, "The latter leads to features that are more robust" => The latter here is k-means whereas the former is k-means++. Shouldn't this have read "The former leads to features that are more robust"?
- On page 15, "We obverse that the model trained on “4r” performs better than the base extractor on k-nn classification on ImageNet." First, "obverse" should read "observe". Second, this should have read "perform worse", as indeed 4r underperforms on k-nn classification.

**Strengths And Weaknesses:**

Strengths
- The problem is well-motivated. Self-supervision is at the heart of modern machine learning systems, and the quality of datasets for pre-training has a significant impact on self-supervised learning. Manual curation is a costly and cumbersome process, so automated pipelines can be particularly helpful.
- The proposed pipeline is interesting. The main idea is quite simple, which can be a plus for the complex and computationally expensive process of data curation. Despite the apparent simplicity, the framework contains interesting novel elements, namely, the resampling -clustering step as well as the hierarchical sampling in a top-down manner. Overall, the framework is interesting and effective.
- The algorithm is justified theoretically, as the authors are able to show that through hierarchical and successive k-means the cluster centroids tend to be distributed uniformly over the support of the original data. The authors additionally provide ample empirical evidence about their theoretical claims, and show how the proposed framework works as intended.
- The empirical evaluation is extensive and mostly confirms the authors' claims. The experiments on synthetic data demonstrate that the framework can indeed sample uniformly from the data support. The ablation study is detailed and confirms the various design choices in the proposed pipeline. Furthermore, the results show that the embeddings from the curated datasets are particularly robust and can handle tasks on which they were not trained.
- Generally well written paper with good citation coverage, even though there are a few typos here and there (see requested changes section).

Weaknesses
- The authors posit that good pre-training datasets for self-supervision must be balanced (besides being large and diverse), and this is indeed what their framework is mostly about. Even though this is quite an obvious fact for several supervised ML tasks (such as ImageNet classification), I do not think it is equally obvious for general self-supervision. In general self-supervision, some concepts may inherently appear more frequently than others, and in fact we may want systems to reflect this. For instance, an LLM that is used for text generation and is pre-trained through self-supervision may need to capture that certain documents/concepts/topics are more frequent and must appear more prominently in the generated text. The authors argue that self-supervision must be balanced by showing results on ImageNet classification (a supervised task) using self-supervised features trained on ImageNet vs. its unbalanced variants. I agree that in the context of supervised image classification, this makes perfect sense. And it may be equally valid in numerous other contexts. But I feel the authors do not provide clear and unambiguous evidence that balanced datasets are universally desired for self-supervision.
- The authors correctly point out that distortion functions with $s>0$ are much harder to deal with, because the computation of a cluster's centroid is not trivial (unlike the case when $s=2$). It is known for example that k-medoids is NP-hard to solve exactly; however, heuristics and approximate solutions exist. It would have been great if the authors had provided results against this non-hierarchical approach (using heuristics or other fast approximate algorithms), so that computation is not an issue (perhaps in a smaller scale setting or synthetic datasets). If, for instance, the proposed framework significantly outperforms other distortion functions, this would provide even stronger evidence in favor of hierarchical k-means.
- The proposed framework critically depends on the embeddings that the hierarchical k-means algorithm makes use of. The authors assume that we can use embeddings produced by some feature extractor, e.g., DINOv2 for images or SBERT for text. Even though we intuitively expect that these embeddings significantly affect the proposed framework, I feel that the authors do not discuss this point in detail, except for a relatively short discussion in Table 3f. The big problem in my view is that these embeddings are used to define the clusters/concepts. Ideally, we would want embeddings that can cover the whole concept space sufficiently well, but is this enough? For instance, the raw data pool performs much worse than 4r in Table 3f, despite the fact that the raw data pool should cover a large variety of concepts. Is this because the tasks in Table 3f are specifically tailored for ImageNet, or are there other reasons? I feel this is a point the authors should investigate in more detail. What for instance makes a certain set of embeddings more preferrable?
- In the image experiments in Table 5, the proposed framework outperforms raw on all classification tasks. But raw performs quite strongly on retrieval tasks, segmentation and depth estimation tasks, and even on the fairness criteria. Overall, I feel that the proposed framework can be more robust, especially in settings involving out of distribution tasks. But raw is a very decent competitor in many other settings. Could this suggest that the proposed framework is more meaningful in specific situations, such as those involving OOD or long-tailed tasks?

---

> ### Author Response · Authors · 2024-07-10
> **Answers to reviewer Q5bR 1/2**
>
> We would like to thank reviewer Q5bR for the valuable remarks and suggestions. We also thank the reviewer for finding the typos. We have corrected them in the manuscript. We provide answers to the reviewer's questions below. We have also updated the manuscript and highlighted the text related to our answers in blue.
>
> *Q.  Can the authors provide stronger evidence that balanced datasets are always preferable for self-supervision? If not, instead of making such a big and universal claim, could the authors elucidate those settings where balanced pre-training datasets are beneficial? For instance, in the context of OOD classification, it is reasonable to claim that balanced datasets would be preferrable. Instead of treating the claim as obvious, it would be helpful if the authors pinpointed various conditions under which balanced pre-training datasets can improve performance (with reasonable certainty), if universal arguments are much harder to write down clearly.*
>
> The main focus of our work is self-supervised representation learning which aims to learn data embeddings that perform universally well on different downstream tasks and data domains, without knowing beforehand those tasks and domains. Intuitively, such a universal representation should avoid biases to any specific data concepts. We make the hypothesis that avoiding these biases requires balanced SSL pre-training datasets. Training on these datasets would result in features that better separate the concepts and an embedding space with a more meaningful distance. By balancing datasets, we do not seek to improve features’ performance on specific individual downstream tasks or data domains, but generally on all tasks and domains. We have validated our hypothesis via extensive experiments. We observe that on average, balanced datasets lead to significantly better performance than the raw data pool on all tasks and domains. Naturally, down-weighting certain head concepts in favor of making tail concepts more prominent would lead to a performance drop in tasks involving these head concepts. This phenomenon is observed with segmentation and depth benchmarks where head concepts such as “building”, “bedroom” and “indoor scene” are relevant. It is noteworthy that the drop in these benchmarks (1.2% relatively on average) is much smaller than the gain observed in other benchmarks such as ImageNet-A (14.6%),  iNaturalist18 (11.6%), ImageNet knn (4.5%) or ImageNet V2 (2.8%).
>
> We agree with the reviewer that balanced datasets are not universally necessary for all self-supervised tasks, e.g. when one wants to learn the distribution of the raw data pool. However, in the particular case of LLMs that the reviewer mentioned, we show via experiments in Sec. 4.3.1 that balanced datasets are also useful. This could be explained by the fact that though often leveraging web-based text data, LLMs never seek to learn their distribution, but rather an unknown language distribution. On the other hand, web-based text data are often noisy and unclean, and their distribution could be very far from the true language distribution. Sampling balanced datasets from this data source could correct this distribution and better approach the language distribution. Other than LLMs, we believe that balancing data is also helpful in other practical applications that leverage large web-based data collections.
>
> *Q. Can the authors provide any empirical data with different distortion functions (with
> )? If computational overhead is a problem, what if they simply experimented with smaller synthetic datasets? This many not be critical, but I feel it would considerably strengthen the work, if the authors could show that alternative frameworks (e.g., based on k-medoids) perform worse than the proposed hierarchical k-means.*
>
> Following the reviewer’s suggestion, we have applied $k$-means with distortion function $\|x-y\|^s$ with $s \in \{4, 64, 256\}$ on the 2D simulated dataset in Sec. 4.1. As expected, following our analysis in Sec. 3.2, we observe that increasing $s$ leads to more uniformly distributed clusters. However, its effect is weaker than that of hierarchical $k$-means: Even with $s=256$, the distribution of the centroids of $k$-means is still further away from the uniform distribution than the distribution of the centroids given by our 3-level hierarchical $k$-means. Furthermore, we have observed numerical instability when running $k$-means with large values of $s$. It is also noteworthy that hierarchical $k$-means enables our hierarchical sampling algorithm, which is shown to bring significant performance gains on downstream tasks, while $k$-means with another distortion function does not. We have added this analysis in Sec. 4.1.

---

> ### Author Response · Authors · 2024-07-10
> **Answers to reviewer Q5bR 2/2**
>
> *Q. Can the authors elucidate the role of embeddings? I feel the discussion in Table 3f is quite elementary. I think it is fair to say that the performance of the framework critically depends on the embeddings. How so? For instance, why 4r-raw performs worse than 4r, even though the former should cover a great deal of concepts in the embedding space? Maybe one would have expected that 4r-raw would be able to sample uniformly from a very large concept space, but in practice it performs worse. Is this expected and how exactly?*
>
> The base embeddings play an important role in our framework since they define the concepts via their Euclidean distance, as pointed out by the reviewer. We give an example of an idea embedding space for our framework in Sec. 3.1: a space that organizes concepts in separate, small blobs of equal-size. The induced Euclidean distance between data points in this space would perfectly reflect human perception: Data points of the same concepts stay closer than data points of different concepts. Sampling from the uniform distribution over the data support in the space would asymptotically result in balanced datasets. Though such embeddings do not exist in practice for real data such as web-based images or text to the best of our knowledge, we still can employ existing embeddings that induce a meaningful distance function. Therefore, we use DINOv2 trained on the balanced, but small ImageNet1k dataset as our base embeddings for images, and SBert embeddings for text. Distance between embeddings from these models has been successfully used for non-parametric image classification and instance retrieval [1] or semantic textual similarity tasks [2].
>
> In Tab. 3f, we compare different types of base image embeddings and show that embeddings trained on ImageNet1k are more suitable to our pipeline than embeddings trained on the raw data pool. This is somewhat expected since the latter induce worse distance function, evidenced by worse performance in ImageNet knn and retrieval (Tab. 3a,f). We believe this is due to the imbalanced nature of the raw data pool. As discussed in Sec. 3.1, an imbalanced dataset could result in SSL features with biases toward the head concepts. Indeed, clustering-based SSL training methods such as DINOv2 divide training batches into roughly equal-sized clusters, and pull training samples in the same clusters closer while pushing samples in different clusters away. When the training dataset is highly imbalanced, training batches are dominated by head concepts, and samples from the same head concept could be divided into different clusters. As such, training samples of the same head concepts are pushed away from each other while there are not enough samples from tail concepts to learn from. Consequently, the induced distance is distorted to distinguish better samples from a few head concepts while discerning less correctly the tail concepts.
>
> *Q. In the image experiments, raw performs very well on dense tasks and even on the fairness criteria. Does this suggest that the proposed framework is mostly expected to shine in OOD and long-tailed tasks, but not more generally? Some discussion there could be quite illuminating.*
>
> We discussed this point in Sec. 4.2.3. The raw’s good performance in these dense tasks is likely due to the prevalence of the “bedroom”, “indoor scene” or “building” concepts which are relevant in benchmarks such as ADE20K, NYU or SUN-RGBD. We looked at the biggest clusters in the highest-level clusters and observed that these concepts take up about 5% of our raw data pool while they only account for about 0.6% of the curated dataset. As discussed above, down-weighting head concepts through balancing could harm the pre-trained features’ performance on downstream tasks involving these concepts. However, we observe that the small drops in these benchmarks are more than compensated with large gains on other benchmarks.
>
> On fairness benchmarks, the curated dataset “4r” leads to better performance than “raw” in 5/7 cases. On average, features trained on the former perform slightly better than those trained on the latter both when considering income level and geographic area. However, we would like to stress that the metric of interest in these benchmarks is the performance discrepancy between different income levels and geographical regions, not the absolute accuracy. We observe that the curated dataset leads to significantly smaller performance discrepancies than the raw data pool.
>
> [1] Maxime Oquab, Timothée Darcet, Théo Moutakanni, Huy V. Vo, Marc Szafraniec, Vasil Khalidov, Pierre Fernandez, Daniel Haziza, Francisco Massa, Alaaeldin El-Nouby, et al. Dinov2: Learning robust visual features without supervision. In TMLR, 2023.
>
> [2] Nils Reimers and Iryna Gurevych. Sentence-bert: Sentence embeddings using siamese bert-networks. In EMNLP, 2019.

---

### Decision · Action_Editor_JevZ · 2024-07-27

**Recommendation:** Accept as is

**Comment:**

Overall, the paper considers a relevant and timely problem. The proposed solution to address concept imbalance through automatic dataset curation is experimentally proven to be effective and this is convincingly supported by experiments both on toy and a variety of realistic datasets. The paper provides significant results that are of interest to the TMLR audience. All reviewers appreciated the work and voted for its acceptance. Based on these considerations, I recommend the acceptance of the paper in its current form.

One important remark is that nothing is mentioned regarding code release. I encourage the authors to provide their code for open and reproducible research.

In the final version, please also consider to correct some typos, for instance:
- respecitvely => respectively
- learned presentation => learned representation
- compact in => compact set in
- points points => points

**Audience:**

The work can attract interest especially in the community of self-supervised learning. More broadly, it can potentially spark some interest in areas related to automated machine learning and fairness.

**Claims And Evidence:**

The paper addresses the general problem of self-supervised learning under concept-imbalance by proposing an algorithm for data selection to rebalance the concept distribution. Experiments on datasets from diverse domains (image, text, remote sensed image data) provide convincing evidence supporting the need for dataset curation and the development of corresponding automated balancing strategies.

Reviewers have appreciated the originality of the proposed solution based on the iterative application of clustering and data subsampling, its theoretical justification with the asymptotic convergence guarantee towards uniformly distributed concepts and corresponding balanced sampling and the thorough experimental analysis. Furthermore, the authors have convincingly addressed all concerns related to clarity and the significance of the results raised by reviewers during the discussion phase, thereby improving the overall quality of the manuscript.